# Foresight Diffusion: Improving Sampling Consistency in Predictive Diffusion Models

**Yu Zhang,**\* **Xingzhuo Guo,**\* **Haoran Xu, Jialong Wu, Mingsheng Long**✉
School of Software, BNRist, Tsinghua University, Beijing 100084, China
`{zhangyu24,gxz23}@mails.tsinghua.edu.cn, mingsheng@tsinghua.edu.cn`

## Abstract

Diffusion and flow-based models have enabled significant progress in generation tasks across various modalities and have recently found applications in predictive learning. However, unlike typical generation tasks that encourage sample diversity, predictive learning entails different sources of stochasticity and requires sampling consistency aligned with the ground-truth trajectory, which is a limitation we empirically observe in diffusion models. We argue that a key bottleneck in learning sampling-consistent predictive diffusion models lies in suboptimal predictive ability, which we attribute to the entanglement of condition understanding and target denoising within shared architectures and co-training schemes. To address this, we propose *Foresight Diffusion (ForeDiff)*, a framework for predictive diffusion models that improves sampling consistency by decoupling condition understanding from target denoising. ForeDiff incorporates a separate deterministic predictive stream to process conditioning inputs independently of the denoising stream, and further leverages a pretrained predictor to extract informative representations that guide generation. Extensive experiments on robot video prediction and scientific spatiotemporal forecasting show that ForeDiff improves both predictive accuracy and sampling consistency over strong baselines, offering a promising direction for predictive diffusion models.

## 1 Introduction

Diffusion models (Sohl-Dickstein et al., 2015; Song & Ermon, 2019; Ho et al., 2020; Song et al., 2020) and flow-based models (Lipman et al., 2023; Liu et al., 2023; Albergo & Vanden-Eijnden, 2023) are a class of generative models that has achieved state-of-the-art across a wide range of tasks and modalities, including image (Dhariwal & Nichol, 2021), video (Ho et al., 2022), and cross-modal generation (Saharia et al., 2022; Singer et al., 2022; Rombach et al., 2022). Owing to their ability to model complex and multimodal distributions, diffusion models have recently been adopted for predictive learning (Voleti et al., 2022; Chen et al., 2023; Gao et al., 2023) within the conditional generation framework, where they serve as spatiotemporal predictors to learn real-world dynamics and generate future trajectories conditioned on past observations.

Although both require high-fidelity stochastic outputs, predictive learning fundamentally differs from generative tasks in the nature of stochasticity. In generative tasks (e.g., text-to-image synthesis), the target distribution corresponding to a certain text prompt is inherently diverse, and models are designed to pursue diversity, allowing imperfect or widely varying samples. In contrast, predictive learning (e.g., robot video prediction) aims to infer physically coherent futures from partial observations, where stochasticity stems mainly from incomplete or partial observational information. Thus, predictive models, though require stochastic outcomes, prioritize per-sample accuracy and therefore should ensure *sampling consistency* to a certain extent,

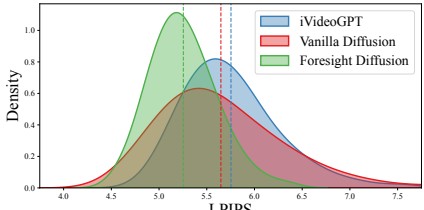

Figure 1: Kernel density estimation: LPIPS distributions of generated samples. Shaded areas represent estimated probability densities; dashed lines indicate sample means. A lower LPIPS corresponds to better quality.

---

\*Equal contribution.

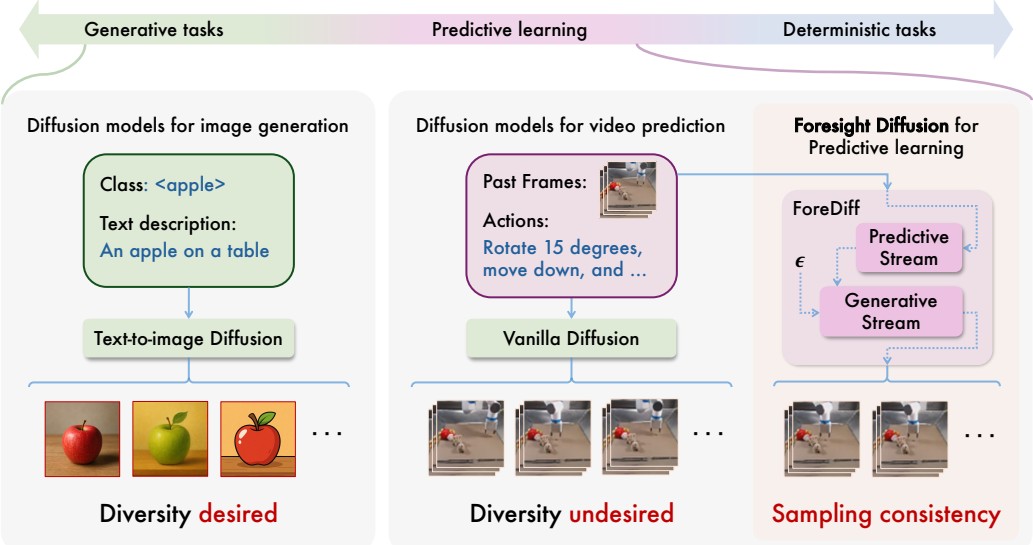

Figure 2: Aligning model stochasticity with task demands. *(Left)* Generative tasks usually favor diversity, making diffusion models ideal as they produce varied samples. *(Middle)* In contrast, predictive learning requires relatively precise predictions, and vanilla diffusion models demonstrate unsatisfactory sampling consistency. *(Right)* Foresight Diffusion strikes a balance between highly stochastic and fully deterministic models, making it well-suited for predictive learning.

which refers to the ability to produce concentrated, low-variance samples under identical conditions such that all the generated samples could closely align with ground-truth trajectories.

The specific demand of sampling consistency challenges the employment of diffusion models to predictive learning, which are prone to issues such as hallucinations, weak conditioning, and sample imperfection (Aithal et al., 2024; Dhariwal & Nichol, 2021). As illustrated in Figure 1, although vanilla diffusion models outperform traditional auto-regressive models in terms of best-case as well as average performance, they exhibit higher sample variance and heavier worst-case tails that are undesirable for predictive learning. This highlights a fundamental mismatch between vanilla diffusion models and the requirements of predictive learning, and consequently there remains a need for sampling-consistent diffusion models that effectively balance stochasticity and determinism.

To address these challenges, we propose *Foresight Diffusion (ForeDiff)*, a framework designed to enhance the sampling consistency of predictive diffusion models by decoupling condition understanding from the denoising process. Diffusion models typically exhibit suboptimal predictive ability compared to deterministic approaches, as condition understanding and target denoising are entangled within shared architectures and co-training schemes, which impairs effective condition understanding and may finally lead to diverse but unfaithful samples relative to the ground truth. Instead of directly applying a conventional conditional diffusion model, ForeDiff introduces a separate deterministic stream that processes condition inputs independently of the stochastic denoising stream. Furthermore, it leverages a pretrained deterministic predictor to extract informative representations, thereby improving the model's predictive ability. Experiments across multiple modalities demonstrate that ForeDiff significantly improves both predictive performance and sampling consistency.

Our contributions are summarized as follows:

- We revisit diffusion models in the context of predictive learning, a task intrinsically different from conventional content generation, and identify their sampling consistency issues.
- We attribute the consistency issues of diffusion models to the entanglement of condition understanding and target denoising within shared architectures and co-training schemes.
- We propose Foresight Diffusion, a framework that improves sampling consistency by decoupling condition understanding and incorporating a pretrained deterministic predictor.
- Extensive experiments and analyses on video prediction and spatiotemporal forecasting demonstrate that ForeDiff achieves superior predictive accuracy and sampling consistency.

## 2 PRELIMINARIES

Denoising-based generative models typically build on a forward process that progressively corrupts clean data samples with increasing amounts of noise. By default, we consider the simplest linear interpolation scheme (Lipman et al., 2023; Liu et al., 2023) which has been widely used due to its analytical tractability and connections to optimal transport. The forward process is expressed as the linear interpolation between a clean sample $\mathbf{x}_0 \sim q(\mathbf{x})$ and standard Gaussian noise $\boldsymbol{\epsilon} \sim \mathcal{N}(\mathbf{0}, \mathbf{I})$:

$$\mathbf{x}_t = (1 - t)\mathbf{x}_0 + t\boldsymbol{\epsilon}, \quad t \in [0, 1] \tag{1}$$

To recover the data distribution, we learn to reverse this process by training a neural network $\mathbf{v}_\theta(\mathbf{x}_t, t)$ to approximate the time-dependent velocity field (or its reparameterizations). The training objective, known as conditional flow matching, is formulated as:

$$\mathcal{L}_{\text{velocity}}(\theta) := \mathbb{E}_{\mathbf{x}_0, \boldsymbol{\epsilon}, t} \left[ \|\mathbf{v}_\theta(\mathbf{x}_t, t) - (\boldsymbol{\epsilon} - \mathbf{x}_0)\|^2 \right]. \tag{2}$$

At inference time, samples are generated by integrating the learned velocity field backward in time:

$$\mathbf{x}_{t-\Delta t} = \mathbf{x}_t - \mathbf{v}_\theta(\mathbf{x}_t, t)\Delta t, \tag{3}$$

starting from a Gaussian noise sample $\mathbf{x}_1 \sim \mathcal{N}(\mathbf{0}, \mathbf{I})$ and progressing toward $t = 0$. This framework naturally extends to conditional generation by introducing conditioning variables $\mathbf{c}$ into the model, resulting in a conditional velocity field $\mathbf{v}_\theta(\mathbf{x}_t, t, \mathbf{c})$, where $\mathbf{c}$ guides the generative dynamics.

## 3 METHOD

### 3.1 DIFFUSION MODELS FOR PREDICTIVE LEARNING

Predictive learning is fundamentally a stochastic task. We formulate it as a conditional generation problem, where $\mathbf{s}^{-O+1:0}$ denotes a sequence of past visual observations and $\mathbf{c}$ represents a potential perturbation to the environment (e.g., actions or goals) (Agarwal et al., 2025). The objective of a predictive model is to approximate the conditional distribution of future frames $p(\mathbf{s}^{1:S} \mid \mathbf{s}^{-O+1:0}, \mathbf{c})$, which is inherently unknown and must be learned from data.

To reduce computational cost, we adopt the widely used latent diffusion paradigm (Rombach et al., 2022), which compresses the frames into a lower-dimensional latent space using a pretrained autoencoder composed of an encoder $E$ and a decoder $D$. The past and future frames are encoded as $\mathbf{z}^{-O+1:0} = E(\mathbf{s}^{-O+1:0})$ and $\mathbf{z}^{1:S} = E(\mathbf{s}^{1:S})$, respectively. Denoting $\mathbf{x} = \mathbf{z}^{1:S}$ and $\mathbf{y} = \{\mathbf{z}^{-O+1:0}, \mathbf{c}\}$, the learning objective becomes modeling the conditional distribution $p(\mathbf{x}|\mathbf{y})$. We focus on predictive diffusion models, which—as described in Section 2—learn a conditional denoiser $\mathbf{v}_\theta(\mathbf{x}_t, \mathbf{y}, t)$ that takes as input both the condition $\mathbf{y}$ and a noisy version of the target $\mathbf{x}_t$.

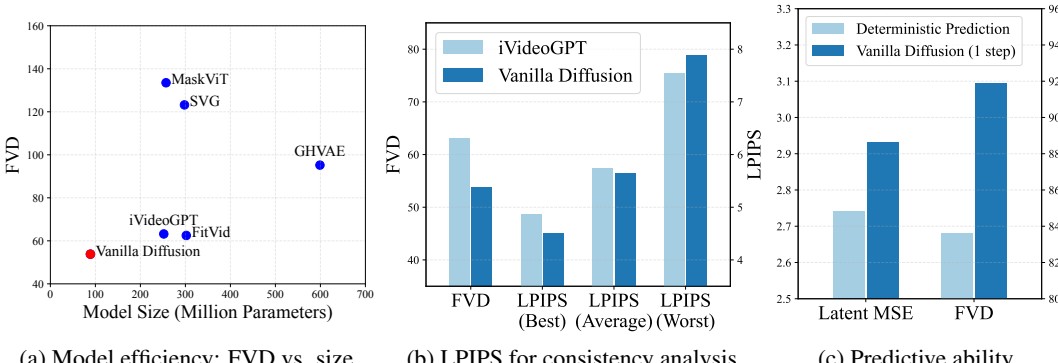

(a) Model efficiency: FVD vs. size.  (b) LPIPS for consistency analysis.  (c) Predictive ability.

Figure 3: Comparison between predictive diffusion models and existing baselines on RoboNet dataset. *(a)* Vanilla diffusion achieves competitive FVD with significantly fewer parameters, demonstrating high model efficiency. *(b)* Vanilla diffusion performs well on best and average LPIPS, but suffers from higher worst-case error, highlighting poor sampling consistency. *(c)* Vanilla diffusion underperforms a deterministic predictor in absence of noisy targets, revealing its limited predictive ability.

**Predictive diffusion models are efficient, accurate, but not consistent.** We evaluate the performance of predictive diffusion models in comparison to existing baselines, including auto-regressive and mask-based architectures (Wu et al., 2024; Gupta et al., 2023). We adopt a video-adapted DiT, shown in Figure 4(a), as our vanilla diffusion model, and assess its performance using two standard metrics: Fréchet Video Distance (FVD) for distributional similarity, and LPIPS for perceptual sample-level quality (Unterthiner et al., 2018; Zhang et al., 2018). Figure 3a presents a scatter plot comparing model size and FVD, while LPIPS scores under best-case, average, and worst-case conditions are reported in Figure 3b. Our key findings are summarized below:

- Despite having a smaller model size and no pretraining, the vanilla diffusion model outperforms baseline models in FVD, best-case LPIPS, and average LPIPS, demonstrating strong efficiency and accuracy for predictive learning.
- However, under worst-case LPIPS, the diffusion model underperforms iVideoGPT. Together with the results in Figure 1, this suggests a lack of sampling consistency, i.e., insufficient concentration of plausible outputs under the same condition.

These results reveal a critical limitation of predictive diffusion models: although exhibiting strong best-case performance with compact architectures, they lack robust conditional control during generation, leading to significant variability across samples. As a result, evaluating models only by best-case performance—as commonly done in prior works (Voleti et al., 2022; Yu et al., 2023; Wu et al., 2024)—can be misleading. Therefore, we report average metrics throughout the remainder of the paper unless otherwise specified.

## 3.2 Observations

To further explore the sampling inconsistency behavior in predictive diffusion models, we investigate their predictive ability, i.e., how well the model understands condition inputs (e.g., visual observations and actions) and predicts future trajectories based on the task's underlying dynamics. To assess how much the model relies solely on $\mathbf{y}$ during denoising, we consider the special case at $t = 1$ (corresponding to $t = T$ under notations in DDPM (Ho et al., 2020)), where $\mathbf{x}_1 \sim \mathcal{N}(\mathbf{0}, \mathbf{I})$ contains no signal and thus contributes no useful information, which introduces the following lemma[1]:

**Lemma 3.1.** *For a diffusion model as defined in Section 2, by reparameterizing the output as* $\hat{\mathbf{x}}_\theta(\mathbf{x}_t, t, \mathbf{y}) = \mathbf{x}_t - t \cdot \mathbf{v}_\theta(\mathbf{x}_t, t, \mathbf{y})$ *(Karras et al., 2022), an ideal diffusion model at $t = 1$ minimizes:*

$$\mathcal{L}_{\text{pred}}(\theta | t = 1) = \mathbb{E}_{\mathbf{x}_0, \mathbf{y}, \boldsymbol{\epsilon}} \left[ \|\hat{\mathbf{x}}_\theta(\boldsymbol{\epsilon}, 1, \mathbf{y}) - \mathbf{x}_0\|_2^2 \right], \tag{4}$$

*where the noise $\boldsymbol{\epsilon}$ is independent of both $\mathbf{x}_0$ and $\mathbf{y}$. Furthermore, a diffusion model minimizing Eq. (4) reduces to a similar architectured deterministic predictor $f_\xi$ minimizing the following objective:*

$$\mathcal{L}_{\text{deter}}(\xi) = \mathbb{E}_{\mathbf{x}_0, \mathbf{y}} \left[ \|f_\xi(\mathbf{y}) - \mathbf{x}_0\|_2^2 \right]. \tag{5}$$

This lemma conveys two insights. First, evaluating performance at $t = 1$ provides a reasonable proxy for assessing the predictive ability of a diffusion model, as the model must rely entirely on $\mathbf{y}$ while discarding the irrelevant input $\boldsymbol{\epsilon}$. Second, the predictive ability of a diffusion model is inherently bounded and is possible to be equivalent to that of a deterministic model. To empirically test whether the diffusion model can reach this upper bound, we train a ViT-based deterministic model $f_\xi$, shown in Figure 4(b), with an architecture analogous to the diffusion model. We then compare its performance to the diffusion model evaluated with a single-step inference at $t = 1$. The results, shown in Figure 3c, reveal that the diffusion model underperforms its deterministic counterpart in both latent and pixel space, verifying its limited predictive ability relative to its potential.

**Discussion.** Such observation indicates that the suboptimal predictive ability of diffusion models stems from the entanglement between condition understanding and target denoising due to the nature of diffusion models to train on various $t$. This entanglement constrains condition understanding by factors of both architecture and training. From the architecture perspective, network parameters must simultaneously learn both condition understanding of $\mathbf{y}$ and target denoising of $\mathbf{x}_t$, and this dual-role

---

[1]Proof is referred to Appendix A.

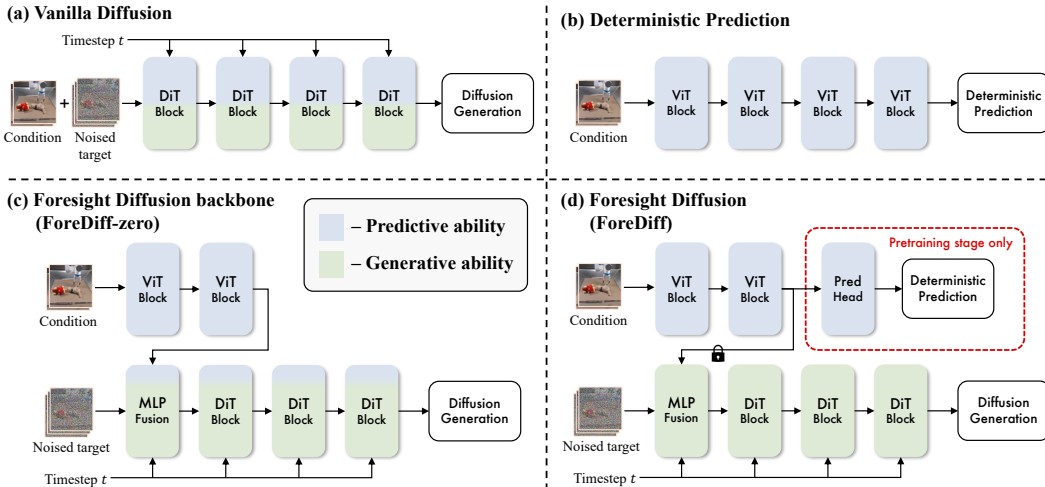

Figure 4: Overview of Foresight Diffusion. *(a)* Vanilla diffusion jointly processes condition and noisy target, limiting its predictive ability. *(b)* A Deterministic model focuses solely on condition understanding and achieves better predictive performance. *(c)* ForeDiff-zero introduces a separate predictive stream to isolate condition understanding from noise. *(d)* ForeDiff further adopts a two-stage scheme: it pre-trains the predictive stream, then freezes its representations to guide generation.

constraint can limit the model's ability to fully exploit the condition information. From the training perspective, the presence of $\mathbf{x}_t$ as an informative input introduces a shortcut, making it easier for the model to rely on generative priors from $\mathbf{x}_t$ rather than precise task-specific dynamics from $\mathbf{y}$.

## 3.3 FORESIGHT DIFFUSION

Previous observations suggest that shared architectures and co-training schemes of condition $\mathbf{y}$ and noisy target $\mathbf{x}_t$ limit the predictive ability of diffusion models, due to the need to simultaneously balance condition understanding and target denoising. To address these limitations, we introduce a simple yet effective framework that enhances predictive ability through architectural decoupling and improved training scheme.

**Architecture.** Building on architectures of vanilla diffusion models, we propose an architectural extension that integrates decoupled deterministic blocks for processing the condition $\mathbf{y}$ independently of $\mathbf{x}_t$. This design forms the *Foresight Diffusion backbone (ForeDiff-zero)*, illustrated in Figure 4(c), which separates the model into two distinct streams, the predictive stream and the generative stream, aiming to focus on $\mathbf{y}$ and $\mathbf{x}_t$ respectively. The predictive stream is composed of deterministic ViT blocks, while the generative stream follows the standard DiT-based denoising process. Formally, let $M$ denote the number of ViT blocks and $N$ the number of DiT blocks. The process can then be formulated as:

$$
\begin{aligned}
\mathbf{g}_0 &= \text{PatchEmbed}(\mathbf{y}), & \mathbf{g}_i &= \text{ViT}_i(\mathbf{g}_{i-1}), \ \ i = 1, \ldots, M, \\
\mathbf{h}_0 &= \text{PatchEmbed}(\mathbf{x}_t), & \mathbf{h}_1 &= \text{Fusion}(\mathbf{h}_0, \mathbf{g}_M, t), \\
\mathbf{h}_{i+1} &= \text{DiT}_i(\mathbf{h}_i, t), \ \ i = 1, \ldots, N, & \hat{\mathbf{v}} &= \text{OutHead}(\mathbf{h}_{N+1}).
\end{aligned}
\tag{6}
$$

When $M = 0$, ForeDiff-zero reduces to vanilla conditional diffusion, where the fusion operates directly on the raw condition: $\mathbf{h}_1 = \text{Fusion}(\mathbf{h}_0, \text{PatchEmbed}(\mathbf{y}), t)$.

Unlike vanilla diffusion models, which ingest both $\mathbf{y}$ and $\mathbf{x}_t$ at the initial point of a shared network, ForeDiff-zero processes the condition solely within its predictive stream and passes the resulting informative representation $\mathbf{g}_M$ instead of $\mathbf{y}$ to the generative stream. Since the predictive stream is entirely agnostic to $\mathbf{x}_t$, its parameters are fully dedicated to understanding $\mathbf{y}$, thereby mitigating the architectural entanglement that limits the predictive ability.

**Training scheme.** To further ensure that the ViT blocks in the predictive stream effectively acquire predictive ability instead of learning static representation, in addition to the end-to-end training

scheme of ForeDiff-zero, we further enhance *Foresight Diffusion (ForeDiff)* by adopting a two-stage training scheme demonstrated in Figure 4(d). In the first stage, the predictive stream is trained as a standalone deterministic predictor. Inspired by the strong predictive ability of the deterministic model $f_\xi$ discussed in Section 3.2, ForeDiff trains the predictive stream by adding a prediction head $\mathrm{PredHead}$ to form the architecture of $f_\xi$, which is defined by $f_\xi(\mathbf{y}) = \mathrm{PredHead}(\mathbf{g}_M)$, and train it using the prediction loss in Eq. (5). In the second stage, we freeze the pretrained predictive stream and remove the $\mathrm{PredHead}$ module. The resulting internal representation $\mathbf{g}_M$, computed by the frozen ViT blocks, is then used as the conditioning input to train the generative stream.

We treat the predictive and generative streams as two independent models, denoted by $P_\xi$ and $G_\theta$, respectively. Each stage is optimized with one of the following loss functions:

$$
\begin{aligned}
\mathcal{L}_{\mathrm{deter}} &= \mathbb{E}_{\mathbf{x}_0, \mathbf{y}} \left[ \left\| P_\xi(\mathbf{y}) - \mathbf{x}_0 \right\|_2^2 \right], \\
\mathcal{L}_{\mathrm{denoise}} &= \mathbb{E}_{\mathbf{x}_0, \mathbf{y}, \boldsymbol{\epsilon}, t} \left[ \left\| G_\theta(\mathbf{x}_t, P'_\xi(\mathbf{y}), t) - (\boldsymbol{\epsilon} - \mathbf{x}_0) \right\|_2^2 \right],
\end{aligned}
\tag{7}
$$

where $P'_\xi$ refers to the predictive stream excluding the $\mathrm{PredHead}$ module. This ensures that the information guiding generation are derived from the learned predictive representations, rather than the final outputs of the predictor. By combining architectural decoupling with dedicated predictive pretraining, ForeDiff leverages a deterministic predictor as a preparatory module for conditional generation. The design enables the model to *"foresee"* contextually rich representations, thereby enhancing predictive ability and improving both generation accuracy and sampling consistency.

## 4 EXPERIMENTS

We evaluate Foresight Diffusion in comparison to conventional conditional diffusion baselines across a range of tasks, covering both (action-conditioned) robot video prediction and scientific spatiotemporal forecasting. We adopt the same model architecture across all tasks, with the only difference being the condition information provided to the model. The model components follow the standard configurations of ViT (Dosovitskiy et al., 2020) and DiT (Peebles & Xie, 2023) (or SiT (Ma et al., 2024)) blocks. Unless otherwise specified, we use 6 ViT blocks in the predictive stream and 12 DiT blocks in the generative stream. Additional implementation details are provided in Appendix B.

### 4.1 ROBOT VIDEO PREDICTION

**Setup.** We begin by evaluating on RoboNet (Dasari et al., 2019) and RT-1 (Brohan et al., 2022), two real-world video datasets widely used for assessing general video prediction performance. RoboNet contains 162k videos collected from 7 robots operating in diverse environments. Following previous works (Babaeizadeh et al., 2021; Wu et al., 2024), the task is to predict 10 future frames given 2 past frames together with actions. RT-1 consists of 87k videos from 13 robots performing hundreds of real-world tasks. Here, the objective is to predict 14 future frames conditioned on 2 past frames and the corresponding instructions. All video frames are resized to $64 \times 64$ pixels for both datasets. We evaluate model performance using widely adopted metrics including FVD (Unterthiner et al., 2018), PSNR (Huynh-Thu & Ghanbari, 2008), SSIM (Wang et al., 2004), and LPIPS (Zhang et al., 2018). In addition, we introduce $\mathrm{STD}_{\mathrm{PSNR}}$, $\mathrm{STD}_{\mathrm{SSIM}}$, and $\mathrm{STD}_{\mathrm{LPIPS}}$, defined as the standard deviation of metric values across multiple generated samples, to numerically represent sampling consistency, where smaller values indicate more consistent predictions. See Appendix C for computation details.

Table 1: Robot video prediction results on RoboNet and RT-1 datasets. SSIM and LPIPS scores are scaled by 100 for convenient display.

| Dataset | Method | FVD ↓ | PSNR ↑ | SSIM ↑ | LPIPS ↓ | $\mathrm{STD}_{\mathrm{PSNR}}$ ↓ | $\mathrm{STD}_{\mathrm{SSIM}}$ ↓ | $\mathrm{STD}_{\mathrm{LPIPS}}$ ↓ |
|---------|--------|-------|--------|--------|---------|---------|---------|---------|
| RoboNet | Vanilla Diffusion | 53.8 | 27.1 | 88.2 | 5.65 | 0.66 | 1.33 | 0.65 |
|  | ForeDiff-zero | 52.7 | 27.2 | 88.4 | 5.54 | 0.68 | 1.36 | 0.66 |
|  | ForeDiff | **51.5** | **27.4** | **88.8** | **5.25** | **0.37** | **0.70** | **0.35** |
| RT-1 | Vanilla Diffusion | 11.7 | 30.4 | 93.6 | 3.79 | 0.97 | 1.11 | 0.53 |
|  | ForeDiff-zero | **11.1** | 30.7 | 93.9 | 3.60 | 0.95 | 1.03 | 0.50 |
|  | ForeDiff | 12.0 | **31.2** | **94.4** | **3.42** | **0.38** | **0.33** | **0.17** |

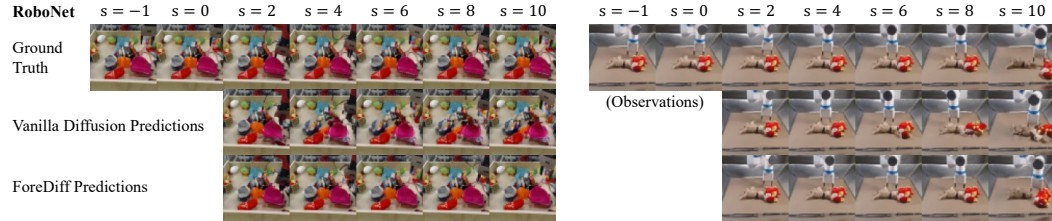

Figure 5: Visualization of results on RoboNet dataset (zoom in for details). In vanilla diffusion models, the pink shovel (left) appears distorted, while the toy object (right) collapses entirely. In contrast, ForeDiff produces more structurally plausible and visually coherent outputs.

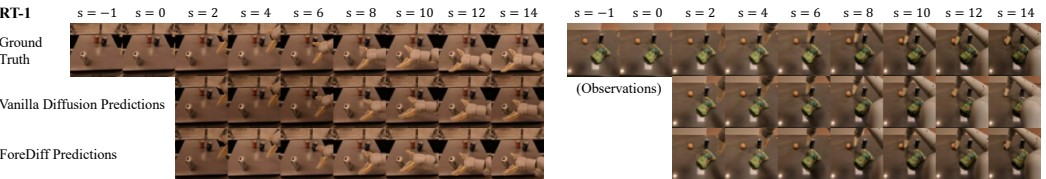

Figure 6: Visualization of results on RT-1 dataset (zoom in for details). Compared with vanilla diffusion, ForeDiff more accurately predicts the brightness of background and the position of robots.

**Results.** We present the experimental results in Table 1. ForeDiff outperforms vanilla diffusion in each dataset, showing not only an improvement in accuracy (PSNR, LPIPS, etc.) but a significant reduction in standard deviation (STD), which demonstrates its suitability for sampling-consistent predictive learning. Moreover, there is no notable difference between ForeDiff-Zero and vanilla diffusion in terms of STD, suggesting that the improved sampling consistency primarily stems from the deterministic pretraining procedure. The qualitative comparisons in Figures 5 and 6 further highlight the superior predictive ability of ForeDiff.

Since post-training strategies such as classifier-free guidance (CFG) (Ho & Salimans, 2022) can also improve performance and consistency, we investigate their relationship with our approach. Specifically, we evaluate the effect of applying CFG during inference to both vanilla diffusion and ForeDiff, shown in 2. Our observations are twofold: (1) the improvement brought by CFG on vanilla diffusion remains limited compared to using ForeDiff, and (2) ForeDiff operates orthogonally to CFG and can be effectively combined with it.

Table 2: Robot video prediction results with CFG on RoboNet.

| Model | Metric | w/o | w/ CFG |
|---|---|---|---|
| Vanilla Diffusion | LPIPS | 5.65 | 5.27 |
| ForeDiff | LPIPS | 5.25 | **5.05** |
| Vanilla Diffusion | STD$_{\text{LPIPS}}$ | 0.65 | 0.49 |
| ForeDiff | STD$_{\text{LPIPS}}$ | 0.35 | **0.24** |

Finally, we compare ForeDiff with prior methods using their evaluation settings, where metrics are reported on the best of 100 samples (Top-1). As shown in Table 3, ForeDiff achieves competitive performance even without accounting for its advantages in consistency, further confirming its effectiveness in predictive learning.

## 4.2 Scientific Spatiotemporal Forecasting

**Setup.** We then evaluate our method on HeterNS (Li et al., 2021; Zhou et al., 2025), which is generated from simulations of heterogeneous 2D Navier-Stokes equations. Each sequence in HeterNS contains 20 frames with a resolution of $64 \times 64$, where each pixel represents the vorticity of turbulence at the corresponding space. The task is to predict the latter 10 frames based on the first 10 frames. L2 and relative L2 are reported according to previous works (Li et al., 2021; Zhou et al., 2025).

**Results.** We present the experimental results in Table 4. ForeDiff achieves a much lower relative L2 error compared to ForeDiff-Zero, and both significantly outperform vanilla diffusion. These results suggest that incorporating deterministic blocks and pre-training them individually contribute to improved performance, demonstrating the applicability of ForeDiff for physical scenarios. Figure 7 illustrates the qualitative advantage of ForeDiff through visual comparisons.

Table 3: Addition results compared with baselines on RoboNet dataset. Metrics are reported on the best of 100 samples. LPIPS and SSIM scores are scaled by 100 for convenient display.

| Method | FVD ↓ | PSNR ↑ | SSIM ↑ | LPIPS ↓ |
|---|---|---|---|---|
| MaskViT (2023) | 133.5 | 23.2 | 80.5 | 4.2 |
| SVG (2019) | 123.2 | 23.9 | 87.8 | 6.0 |
| GHVAE (2021) | 95.2 | 24.7 | 89.1 | 3.6 |
| FitVid (2021) | 62.5 | **28.2** | 89.3 | **2.4** |
| iVideoGPT (2024) | 63.2 | 27.8 | **90.6** | 4.9 |
| ForeDiff | **51.5** | **28.2** | 90.4 | 4.5 |

Table 4: Scientific spatiotemporal forecasting results on HeterNS dataset. Metrics are scaled by 100 for convenient display.

| Method | L2 ↓ | Relative L2 ↓ |
|---|---|---|
| Vanilla Diffusion | 1.73 | 1.50 |
| ForeDiff-zero | 1.03 | 0.83 |
| ForeDiff | **0.19** | **0.18** |

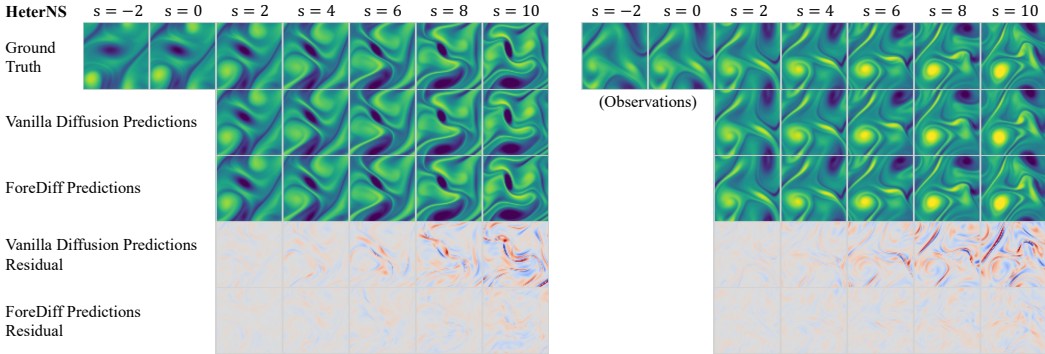

Figure 7: Visualization of results on HeterNS dataset. As the simulation progresses, predictions from vanilla diffusion deviate increasingly, whereas ForeDiff maintains consistently accurate predictions.

## 4.3 ANALYSIS

For clarity, we report simplified results here, focusing on key trends. Complete numerical results across datasets and variants are provided in Appendix D as supplements.

**Effect of** PredHead **module.** We investigate whether ForeDiff benefits more from the predictive ability of the ViT stream, represented by its learned internal features, or its explicit prediction outputs. To this end, we conduct an ablation where the DiT stream is conditioned on PredHead outputs instead of internal representations used in the standard ForeDiff. As shown in Figure 8a, conditioning on PredHead outputs leads to reduced model performance, which supports our hypothesis that the learned predictive representations, rather than the explicit prediction outputs, are more beneficial to the generative process.

**Effect of ViT block number.** To assess the extent of predictive ability required to benefit the generative process, we vary the number of ViT blocks $M$ in the predictive stream from $M = 0$ (i.e., vanilla diffusion) to $M = 12$, while keeping the number of DiT blocks in the generative stream fixed. As shown in Figure 8b, adding a moderate number of ViT blocks noticeably improves performance, but further increasing $M$ yields diminishing gains. This suggests that the predictive ability required to assist a fixed generative backbone can be achieved with minimal overhead, and that even a lightweight deterministic auxiliary module can provide meaningful improvements.

**Effect of design beyond parameter scaling.** We further validate that the performance gains of ForeDiff are attributable to its architectural design rather than the simple scaling of parameters. To isolate the effect of parameter scaling, we extend the vanilla diffusion model to 18 DiT blocks to match the total number of layers used in ForeDiff, and compare its performance with both ForeDiff-Zero and ForeDiff under identical ViT/DiT block configurations. We also evaluate the standalone deterministic predictive stream used in ForeDiff. As shown in Figure 8c, ForeDiff outperforms both the extended vanilla diffusion model and the deterministic stream by a substantial margin. These results indicate that the combination of deterministic prediction and conditional diffusion contributes synergistically, highlighting the effectiveness of the proposed hybrid architecture for predictive learning.

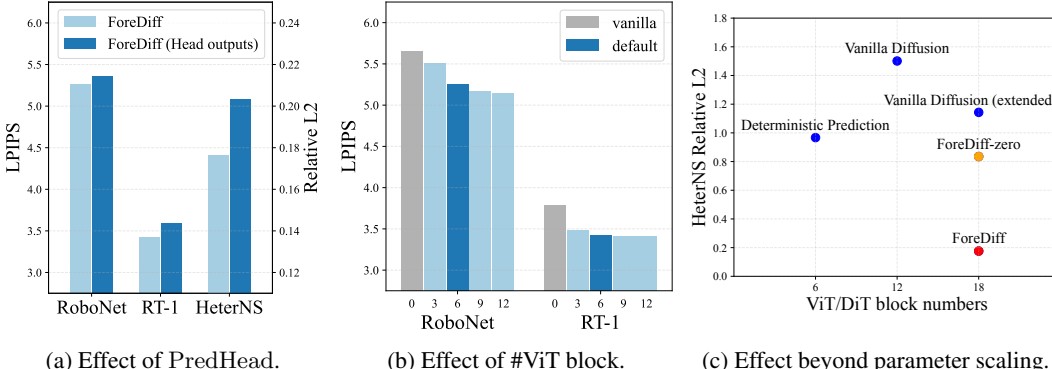

(a) Effect of $\mathrm{PredHead}$.     (b) Effect of #ViT block.     (c) Effect beyond parameter scaling.

Figure 8: Ablation studies. *(a)* Conditioning on $\mathrm{PredHead}$ outputs leads to degraded performance. *(b)* Increasing #ViT block improves accuracy, but further blocks yield diminishing gains. *(c)* ForeDiff outperforms both an extended vanilla diffusion and the standalone deterministic predictor.

# 5 RELATED WORK

**Predictive Learning.** Diffusion models have been widely adopted for predictive learning. Earlier approaches have explored architectural design for incorporating signals within the conditional generation framework, including concatenation- and modulation-based fusion (Voleti et al., 2022; Ho et al., 2022; Blattmann et al., 2023) with U-Net backbones, scalable multi-modal transformer-based architectures (OpenAI, 2024; Yang et al., 2024) and domain-specific architectures (Gao et al., 2023). Beyond diffusion-based approaches, various frameworks have been proposed for predictive tasks, including RNN-based (Shi et al., 2015; Villegas et al., 2019; Wang et al., 2022), auto-regressive (Yan et al., 2021; Wu et al., 2024), and mask-based (Gupta et al., 2023; Yu et al., 2023) methods. As concluded in Section 3.1, diffusion models serve as strong baselines but tend to exhibit greater sample variability compared to these non-diffusion counterparts. This observation motivates our focus on enhancing sampling consistency in predictive diffusion models.

**Diffusion models as components.** While many approaches employ diffusion models in an end-to-end manner, recent studies have integrated them as components within broader architectures. For example, Li et al. (2024) and Liu et al. (2025) incorporate diffusion losses for image mask reconstruction and auto-regressive time-series forecasting, respectively, thereby leveraging the generative ability in a modular way. By comparison, Foresight Diffusion positions the diffusion model as the central component, enhances its consistency within the generation framework while assigning deterministic modules an auxiliary role tailored for condition understanding in prediction tasks.

**Sampling methods.** Beyond standard sampling strategies as Eq. (3), prior work has introduced post-training techniques to address imperfections in diffusion model learning, including classifier (Dhariwal & Nichol, 2021) or classifier-free (Ho & Salimans, 2022) guidance, initial noise manipulation (Ahn et al., 2024; Zhou et al., 2024) and inference-time scaling (Ma et al., 2025). While effective for enhancing sample quality or conditioning, these methods serve as complementary post-hoc techniques alongside existing architectures, operating orthogonally to ForeDiff.

# 6 CONCLUSION

We proposed Foresight Diffusion (ForeDiff), a framework that improves sampling consistency in predictive diffusion models by decoupling condition understanding from target denoising. Through a hybrid architecture that separates predictive and generative processes, and a two-stage training scheme that leverages pretrained deterministic predictors, ForeDiff overcomes key limitations of vanilla diffusion models—particularly their suboptimal predictive ability and high sample variance. Extensive experiments across real-world robot video prediction and scientific forecasting demonstrate that ForeDiff achieves superior accuracy and significantly enhanced sampling consistency, marking a step toward more reliable and controllable predictive diffusion models.

ACKNOWLEDGEMENTS

This work was supported by the National Natural Science Foundation of China (U2342217), the Fundamental and Interdisciplinary Disciplines Breakthrough Plan of the Ministry of Education of China (JYB2025XDXM803), the Beijing Scholar Program, and the National Engineering Research Center for Big Data Software.

We thank our colleagues for their support, especially Yuchen Zhang and Yuezhou Ma for valuable discussions, as well as Haoran Zhang and Huakun Luo for helpful suggestions on figure design.

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

## A  PROOF OF LEMMA

In this section we present the proof of Lemma 3.1.

*Proof.* The proof is structured in three parts: (1) $\hat{\mathbf{x}}$ reparameterization, (2) the $\epsilon$-agnostic property of the optimal solution $\hat{\mathbf{x}}_\theta$, and (3) the reduction to a deterministic prediction model $f_\xi$ in practice.

Starting from definition $\hat{\mathbf{x}}_\theta(\mathbf{x}_t, t, \mathbf{y}) = \mathbf{x}_t - t \cdot \mathbf{v}_\theta(\mathbf{x}_t, t, \mathbf{y})$ with $\mathbf{x}_t = (1-t)\mathbf{x}_0 + t\epsilon$, it holds that

$$
\begin{aligned}
\mathcal{L}_{\text{pred}}(\theta) &= \mathbb{E}_{\mathbf{x}_0, \mathbf{y}, \epsilon, t}\left[\|\hat{\mathbf{v}}_\theta(\mathbf{x}_t, t, \mathbf{y}) - (-\mathbf{x}_0 + \epsilon)\|_2^2\right] \\
&= \mathbb{E}_{\mathbf{x}_0, \mathbf{y}, \epsilon, t}\left[\left\|\frac{\hat{\mathbf{x}}_\theta(\mathbf{x}_t, t, \mathbf{y}) - \mathbf{x}_t}{t} - \frac{\mathbf{x}_0 - \mathbf{x}_t}{t}\right\|_2^2\right] \\
&= \mathbb{E}_{\mathbf{x}_0, \mathbf{y}, \epsilon, t}\left[\frac{1}{t^2}\|\hat{\mathbf{x}}_\theta(\mathbf{x}_t, t, \mathbf{y}) - \mathbf{x}_0\|_2^2\right],
\end{aligned}
$$

and by letting $t = 1$ we arrive

$$
\mathcal{L}_{\text{pred}}(\theta|t=1) = \mathbb{E}_{\mathbf{x}_0, \mathbf{y}, \epsilon}\left[\|\hat{\mathbf{x}}_\theta(\mathbf{x}_1, 1, \mathbf{y}) - \mathbf{x}_0\|_2^2\right] = \mathbb{E}_{\mathbf{x}_0, \mathbf{y}, \epsilon}\left[\|\hat{\mathbf{x}}_\theta(\epsilon, 1, \mathbf{y}) - \mathbf{x}_0\|_2^2\right].
$$

Since $\epsilon$ is independent of both network input $\mathbf{y}$ and target $\mathbf{x}_0$, the involvement of $\epsilon$ contributes no information in the case of $t = 1$, which matches the fact that $\epsilon$ is sampled randomly.

Further, notice the bias-variance decomposition

$$
\begin{aligned}
\mathbb{E}_{\mathbf{x}_0, \mathbf{y}, \epsilon}\left[\|\hat{\mathbf{x}}_\theta(\epsilon, 1, \mathbf{y}) - \mathbf{x}_0\|_2^2\right] = \mathbb{E}_{\mathbf{y}}\Big[&\left\|\mathbb{E}_\epsilon[\hat{\mathbf{x}}_\theta(\epsilon, 1, \mathbf{y})] - \mathbb{E}_{\mathbf{x}_0|\mathbf{y}}[\mathbf{x}_0]\right\|_2^2 \\
&+ \mathbb{D}_\epsilon[\hat{\mathbf{x}}_\theta(\epsilon, 1, \mathbf{y})] + \mathbb{D}_{\mathbf{x}_0|\mathbf{y}}[\mathbf{x}_0]\Big].
\end{aligned}
$$

By the convexity of the $\ell_2$ loss, there exists $\epsilon_0(\mathbf{y})$ satisfying

$$
\left\|\hat{\mathbf{x}}_\theta(\epsilon_0(\mathbf{y}), 1, \mathbf{y}) - \mathbb{E}_{\mathbf{x}_0|\mathbf{y}}[\mathbf{x}_0]\right\|_2^2 \le \left\|\mathbb{E}_\epsilon([\hat{\mathbf{x}}_\theta(\epsilon, 1, \mathbf{y})] - \mathbb{E}_{\mathbf{x}_0|\mathbf{y}}[\mathbf{x}_0]\right\|_2^2.
$$

Therefore,

$$
\begin{aligned}
\mathcal{L}_{\text{pred}}(\theta|t=1) &\ge \mathbb{E}_{\mathbf{y}}\left[\left\|\mathbb{E}_\epsilon[\hat{\mathbf{x}}_\theta(\epsilon, 1, \mathbf{y})] - \mathbb{E}_{\mathbf{x}_0|\mathbf{y}}[\mathbf{x}_0]\right\|_2^2 + \mathbb{D}_{\mathbf{x}_0|\mathbf{y}}[\mathbf{x}_0]\right] \\
&\ge \mathbb{E}_{\mathbf{y}}\left[\left\|\hat{\mathbf{x}}_\theta(\epsilon_0(\mathbf{y}), 1, \mathbf{y}) - \mathbb{E}_{\mathbf{x}_0|\mathbf{y}}[\mathbf{x}_0]\right\|_2^2 + \mathbb{D}_{\mathbf{x}_0|\mathbf{y}}[\mathbf{x}_0]\right] \\
&= \mathbb{E}_{\mathbf{x}_0, \mathbf{y}}\left[\left\|\hat{\mathbf{x}}_\theta(\epsilon_0(\mathbf{y}), 1, \mathbf{y}) - \mathbf{x}_0\right\|_2^2\right],
\end{aligned}
$$

i.e., the loss of diffusion models at $t = 1$ is bounded by that of a certain deterministic model $f(\mathbf{y}) = \hat{\mathbf{x}}_\theta(\epsilon_0(\mathbf{y}), 1, \mathbf{y})$ which minimizes the following objective:

$$
\mathcal{L}_{\text{deter}} = \mathbb{E}_{\mathbf{x}_0, \mathbf{y}}\left[\|f(\mathbf{y}) - \mathbf{x}_0\|_2^2\right].
$$

Finally, we show that the inequalities can be turned into equalities in practical network architectures, which requires $\hat{\mathbf{x}}_\theta(\epsilon_1, 1, \mathbf{y}) = \hat{\mathbf{x}}_\theta(\epsilon_2, 1, \mathbf{y})$ for any $\epsilon_1, \epsilon_2$. Considering the first layer to process $\epsilon$ as $\mathbf{W}\epsilon + \mathbf{b}$, setting $\mathbf{W} = \mathbf{0}$ removes the dependence on $\epsilon$. In this case, with $\mathbf{b}$ and $t = 1$ absorbed into the network parameters, $\hat{\mathbf{x}}_\theta(\epsilon, 1, \mathbf{y})$ reduces to a deterministic model $f_\xi(\mathbf{y})$, completing the proof.

$\square$

## B  IMPLEMENTATION DETAILS

We adopt the widely used latent diffusion paradigm, which operates in a compressed latent space and relies on a pretrained autoencoder consisting of an encoder and a decoder. The autoencoder downsamples the input by a factor of $4 \times 4$ and maps it to a latent representation with 3 channels. All prediction and denoising operations are performed in this latent space. The autoencoder architecture follows the standard design of Stable Diffusion (Rombach et al., 2022), and dataset-specific architectural and training details are summarized in Table 5.

Table 5: Architecture and training details of the pretrained autoencoder for each dataset.

|  | RoboNet | RT-1 | HeterNS |
|---|---|---|---|
| Parameters |  | 55M |  |
| Resolution |  | $64 \times 64$ |  |
| Input channels | 3 | 3 | 1 |
| Latent channels |  | 3 |  |
| Down blocks |  | 3 |  |
| Down layers per block |  | 2 |  |
| Down channels |  | [128, 256, 512] |  |
| Mid block attention |  | True |  |
| Up blocks |  | 3 |  |
| Up layers per block |  | 2 |  |
| Up channels |  | [512, 256, 128] |  |
| Normalization |  | GroupNorm |  |
| Norm groups |  | 32 |  |
| Activation |  | SiLU |  |
| Training steps | $5 \times 10^6$ | $3 \times 10^6$ | $3 \times 10^6$ |
| Discriminator start step |  | $5 \times 10^4$ |  |
| Batch size |  | 16 |  |
| Learning rate |  | $4 \times 10^{-5}$ |  |
| LR schedule |  | Constant |  |
| Optimizer |  | AdamW |  |

Our main model follows the DiT (Peebles & Xie, 2023) and ViT (Dosovitskiy et al., 2020) architecture families, using a unified design across all tasks. We adopt ViT-S and DiT-S configurations for the predictive and generative streams, respectively. Patch embeddings are extracted using a minimal PatchEmbed module that partitions inputs into non-overlapping $2 \times 2$ patches, followed by a linear projection. Sinusoidal positional encodings are used, with embedding dimensions partitioned in a 3:3:2 ratio among the two spatial axes and the temporal axis. Detailed block settings are shown in Table 6. All models are trained using the AdamW optimizer with a constant learning rate of $1 \times 10^{-4}$ and dataset-specific training steps (see Appendix C).

Table 6: Architecture settings for ViT-S and DiT-S blocks used in ForeDiff.

|  | ViT-S (predictive stream) | DiT-S (generative stream) |
|---|---|---|
| Number of blocks | 6 | 12 |
| Hidden size | 384 | 384 |
| MLP ratio | 4.0 | 4.0 |
| Attention heads | 6 | 6 |
| Positional encoding | Sinusoidal | Sinusoidal |
| Normalization | LayerNorm | AdaLN |
| Activation | SiLU | SiLU |

To construct the conditioning input, we follow the masking-based strategy introduced in prior work (Blattmann et al., 2023). Specifically, a temporal binary mask $\mathbf{m} \in \{0, 1\}^{O+S}$ is defined over the full sequence $\mathbf{s}^{-O+1:S}$ to indicate which frames are observed (condition) and which are to be

predicted (target), where $O$ and $S$ denote the numbers of observed and future frames, respectively. After encoding the full sequence into the latent space via $\mathbf{z} = E(\mathbf{s}^{-O+1:S}) \in \mathbb{R}^{(O+S) \times C \times H \times W}$, we apply the mask $\mathbf{m}$ along the temporal axis by computing $\mathbf{z}_{\text{cond}} = \mathbf{m} \cdot \mathbf{z}$, where the mask is broadcast across spatial and channel dimensions. To make the temporal structure explicit, we further concatenate the binary mask itself (channel-wise) to the masked latent $\mathbf{z}_{\text{cond}}$, and feed the result into the predictive stream.

When additional conditioning inputs such as actions or goals $\mathbf{c}$ are available (e.g., in RoboNet and RT-1), we directly concatenate them with the masked latent $\mathbf{z}_{\text{cond}}$ along the channel axis. This enables the model to jointly reason over visual history and auxiliary task-specific signals.

Overall, this masking-based design supports a unified conditioning mechanism for both video-only and video+action tasks. The masked latent input, along with any auxiliary information (e.g., actions), is processed by the predictive stream to produce intermediate features $\mathbf{g}_M$, which serve as conditioning signals for the generative stream.

To inject this condition information into the generative stream, we use a lightweight Fusion module composed of an adaptive layer normalization (AdaLN) layer followed by a two-layer MLP with GELU activation. The fusion operation is defined as:

$$\mathbf{h}_1 = \text{MLP}(\text{AdaLN}([\mathbf{h}_0; \mathbf{g}_M], t)), \tag{8}$$

where $\mathbf{h}_0$ denotes the noisy target, $\mathbf{g}_M$ is the output of the predictive stream (prior to the PredHead module), and $t$ is the diffusion timestep. We first concatenate $\mathbf{h}_0$ and $\mathbf{g}_M$, apply AdaLN conditioned on $t$, and then pass the normalized features through the MLP to obtain the fused representation.

## C    EXPERIMENTAL SETUP

We evaluate ForeDiff on three benchmark datasets covering both real-world and simulated spatiotemporal dynamics: RoboNet (Dasari et al., 2019), RT-1 (Brohan et al., 2022), and HeterNS (Li et al., 2021; Zhou et al., 2025). This section provides additional details on dataset setups, task configurations, and evaluation metrics. All experiments are conducted in a single NVIDIA-A100 40G GPU.

**RoboNet.**    RoboNet is a large-scale real-world video dataset for vision-based robotic manipulation, consisting of approximately 162,000 trajectories collected across seven different robot platforms from four institutions. Each trajectory includes RGB video frames and associated action sequences, all represented in a unified end-effector control space. The dataset captures a wide range of variations in robot embodiment (e.g., Sawyer, Kuka, Franka), gripper design, camera viewpoints, surfaces, and lighting conditions.

Following prior works (Babaeizadeh et al., 2021; Wu et al., 2024), we resize all frames to $64 \times 64$ and predict 10 future frames based on 2 observed frames and corresponding actions. All models are trained on RoboNet for $1 \times 10^6$ steps with a batch size of 16. We report PSNR, SSIM, LPIPS, and FVD as evaluation metrics. In addition, we compute standard deviation (STD) across multiple samples to quantify sampling consistency.

**RT-1.**    RT-1 is a large-scale real-world robotics dataset comprising over 130,000 episodes collected from a fleet of 13 mobile manipulators performing diverse manipulation tasks in office kitchen environments. Each episode includes an RGB video sequence, a natural language instruction, and the executed robot actions. The dataset covers over 700 distinct tasks involving object interaction, long-horizon manipulation, and routine execution.

We formulate the prediction task by conditioning on 2 observed frames and a task instruction, and predicting the next 14 frames. All frames are resized to $64 \times 64$. Training on RT-1 is conducted for $5 \times 10^5$ steps with a batch size of 16, and the evaluation protocol matches that of RoboNet, using PSNR, SSIM, LPIPS, FVD, and STD of these metrics across samples.

**HeterNS.**    HeterNS is a synthetic dataset generated by numerically solving the two-dimensional incompressible Navier–Stokes equations in vorticity formulation over a periodic domain. The

governing system is defined as:

$$\partial_t w(x,t) + u(x,t) \cdot \nabla w(x,t) = \nu \Delta w(x,t) + f(x), \quad x \in (0,1)^2, \; t \in (0,T], \qquad (9a)$$
$$\nabla \cdot u(x,t) = 0, \qquad\qquad x \in (0,1)^2, \; t \in [0,T], \qquad (9b)$$
$$w(x,0) = w_0(x), \qquad\qquad x \in (0,1)^2, \qquad (9c)$$

where $w$ is the vorticity field, $u$ is the divergence-free velocity field recovered via the stream function $\psi$ satisfying $\Delta \psi = -w$, $\nu$ is the viscosity coefficient, and $f(x)$ is a time-invariant external forcing term.

To construct a diverse collection of PDE instances, we vary two key physical parameters: the viscosity $\nu$ and the structure of the forcing term $f(x)$. Specifically, the training set is constructed from a Cartesian grid of parameter configurations:

$$\nu \in \{1 \times 10^{-5}, \; 1 \times 10^{-4}, \; 1 \times 10^{-3}\}, \quad f(x) \in \{5 \text{ distinct variants}\},$$

yielding a total of $3 \times 5 = 15$ unique PDE settings.

Among the forcing terms, three are defined as:

$$f(x) = 0.1 \left( \sin(\omega_1 \pi (x_1 + x_2)) + \cos(\omega_2 \pi (x_1 + x_2)) \right),$$

with frequency pairs $(\omega_1, \omega_2) \in \{(2,2), \; (2,4), \; (4,4)\}$. The fourth variant is defined as:

$$f(x) = 0.1 \left( \sin(2\pi(x_1 - x_2)) + \cos(2\pi(x_1 - x_2)) \right),$$

and the fifth as:

$$f(x) = 0.1 \left( \sin(2\pi(x_1^2 + x_2^2)) - \cos(2\pi(x_1^2 + x_2^2)) \right).$$

These five variants cover a range of spatial patterns, including directional, cross-diagonal, and radially symmetric forcings.

For each of the 15 configurations, we simulate 1000 trajectories, resulting in a total of 15,000 samples. Each trajectory consists of 20 vorticity fields at a spatial resolution of $64 \times 64$. The prediction task involves forecasting the final 10 frames given the first 10 as context. Each model is trained for $5 \times 10^5$ steps using a batch size of 16.

**Evaluation metrics.** We adopt a suite of evaluation metrics to assess model performance across multiple dimensions, including visual quality, perceptual fidelity, predictive accuracy, and sampling consistency. These include PSNR, SSIM, LPIPS, FVD, Relative L2, and STD, described as follows:

- **Peak Signal-to-Noise Ratio (PSNR)** (Huynh-Thu & Ghanbari, 2008) quantifies the ratio between the maximum possible pixel intensity and the distortion introduced by reconstruction errors. It is defined in logarithmic scale (dB) and commonly used to evaluate frame-wise reconstruction quality. A higher PSNR value indicates less pixel-level distortion and better fidelity to the original ground truth frame.

- **Structural Similarity Index Measure (SSIM)** (Wang et al., 2004) measures perceptual similarity by comparing luminance, contrast, and structural information between images. Unlike PSNR, SSIM better correlates with human visual perception. The SSIM score ranges from -1 to 1, where 1 denotes perfect structural similarity. For better readability, we scale SSIM scores by a factor of 100 in all reported results.

- **Learned Perceptual Image Patch Similarity (LPIPS)** (Zhang et al., 2018) evaluates perceptual similarity using deep features extracted from a pretrained neural network (e.g., VGG or AlexNet). It computes the L2 distance between feature representations of two images, capturing differences beyond pixel values. Lower LPIPS values indicate better perceptual quality. Similar to SSIM, LPIPS scores are multiplied by 100 for display.

- **Fréchet Video Distance (FVD)** (Unterthiner et al., 2018) extends the Fréchet Inception Distance (FID) to video generation by considering temporal dynamics. It compares the distributions of real and generated videos in a feature space extracted from a pretrained video recognition model. FVD is sensitive to both spatial quality and temporal consistency, making it a strong indicator of overall generative performance.

- **Relative L2 Distance** (Li et al., 2021) captures normalized regression error in pixel or field-level prediction tasks. It is computed as:

$$\text{Relative L2} = \frac{\|\hat{x} - x\|_2}{\|x\|_2},$$

where $\hat{x}$ and $x$ denote the prediction and ground truth, respectively. A lower relative L2 value implies that the model produces more accurate outputs with respect to both magnitude and structure. This metric is particularly suited for scientific forecasting tasks such as fluid dynamics.

- **Standard Deviation (STD)** of metrics is used to evaluate sampling consistency across multiple generations from the same condition. For each condition input (e.g., past frames and action/instruction), we generate $N$ samples (typically $N = 100$), compute a chosen metric $\mathcal{M}$ (e.g., PSNR, SSIM, LPIPS) for each sample, and calculate the standard deviation of these scores. Let $\mathcal{M}_1^{(i)}, \ldots, \mathcal{M}_N^{(i)}$ denote the metric values for the $i$-th condition, the per-condition standard deviation is:

$$\text{STD}^{(i)} = \sqrt{\frac{1}{N} \sum_{j=1}^{N} \left( \mathcal{M}_j^{(i)} - \bar{\mathcal{M}}^{(i)} \right)^2}, \quad \text{where} \quad \bar{\mathcal{M}}^{(i)} = \frac{1}{N} \sum_{j=1}^{N} \mathcal{M}_j^{(i)}.$$

The final STD score is then averaged over all $C$ conditions:

$$\text{STD} = \frac{1}{C} \sum_{i=1}^{C} \text{STD}^{(i)}.$$

Lower STD indicates that the model produces more consistent outputs across stochastic samples, reflecting stronger reliability under the same input condition.

## D  MORE EXPERIMENTAL RESULTS

This section provides the full numerical results corresponding to the analyses discussed in Section 4.3. While the main paper focuses on reporting key trends and visualizations, the tables below include complete metric values across datasets and variants, serving as a quantitative supplement.

In addition, we include extended experiments covering: (i) the necessity of architectural decoupling, (ii) the necessity of the two-stage training scheme, (iii) the sensitivity of ForeDiff to predictor quality, and (iv) calibration-oriented evaluation (CRPS, NLL, and coverage curves). These results further support the conclusions drawn in the main text.

**Effect of** PredHead **module.**    Table 7 reports the full numerical results for the PredHead ablation across all datasets.

Table 7: Ablation results on the PredHead module. Across datasets, removing PredHead consistently improves both perceptual and pixel-wise metrics. SSIM, LPIPS, L2, and Relative L2 are scaled by 100.

| Dataset | Method | FVD ↓ | PSNR ↑ | SSIM ↑ | LPIPS ↓ |
|---------|--------|-------|--------|--------|---------|
| RoboNet | ForeDiff | 51.5 | 27.4 | 88.8 | 5.25 |
|         | ForeDiff (with PredHead) | 53.7 | 27.3 | 88.7 | 5.35 |
| RT-1 | ForeDiff | 12.0 | 31.2 | 94.4 | 3.42 |
|      | ForeDiff (with PredHead) | 12.4 | 31.0 | 94.1 | 3.60 |

| Dataset | Method | L2 ↓ | Relative L2 ↓ |
|---------|--------|------|---------------|
| HeterNS | ForeDiff | 0.19 | 0.18 |
|         | ForeDiff (with PredHead) | 0.23 | 0.20 |

**Effect of ViT block number.**    Table 8 presents detailed results of varying the number of predictive (ViT) blocks, while keeping the number of generative (DiT) blocks fixed.

Table 8: Effect of varying ViT block number (denoted by $M$) on performance, with DiT blocks fixed to 12. Adding a moderate number of predictive blocks improves performance, while further increases yield diminishing returns. SSIM and LPIPS are scaled by 100.

| Dataset | Method | FVD ↓ | PSNR ↑ | SSIM ↑ | LPIPS ↓ |
|---|---|---|---|---|---|
| RoboNet | M=0 (vanilla) | 53.8 | 27.1 | 88.2 | 5.65 |
| | M=3 | 53.3 | 27.1 | 88.3 | 5.51 |
| | M=6 (default) | 51.5 | 27.4 | 88.8 | 5.25 |
| | M=9 | 50.8 | 27.5 | 89.0 | 5.17 |
| | M=12 | 52.1 | 27.5 | 89.1 | 5.14 |
| RT-1 | M=0 (vanilla) | 11.7 | 30.4 | 93.6 | 3.79 |
| | M=3 | 11.8 | 31.0 | 94.2 | 3.49 |
| | M=6 (default) | 12.0 | 31.2 | 94.4 | 3.42 |
| | M=9 | 12.4 | 31.3 | 94.4 | 3.41 |
| | M=12 | 12.4 | 31.3 | 94.4 | 3.41 |

**Effect of design beyond parameter scaling.** Table 9 provides the complete quantitative results for the ablation study on HeterNS dataset, used to isolate the effect of architectural design from mere parameter scaling.

Table 9: ForeDiff clearly outperforms both deterministic-only and extended vanilla diffusion models, confirming that its improvements stem from architectural design rather than model size alone. Metrics are scaled by 100.

| Method | L2 ↓ | Relative L2 ↓ |
|---|---|---|
| Deterministic Prediction | 1.06 | 0.97 |
| Vanilla Diffusion | 1.73 | 1.50 |
| Vanilla Diffusion (extended) | 1.29 | 1.14 |
| ForeDiff-Zero | 1.03 | 0.83 |
| ForeDiff | 0.19 | 0.18 |

**Necessity of architectural decoupling.** We examine whether training-scheme decoupling alone, without architectural separation, is sufficient to disentangle condition understanding from target denoising. To this end, we first trained a diffusion model exclusively at timestep $t = 1$, enabling it to mimic the behavior of a deterministic predictor (see Lemma 3.1), and then fine-tuned it across all timesteps with uniform weighting. We refer to this variant as *training-only decoupling*, since its architecture remains unchanged while the training procedure is modified through pretraining at $t = 1$ followed by fine-tuning. Table 10 reports the quantitative results on RoboNet. Such pretraining improves sampling consistency but still falls short of ForeDiff in both consistency and overall quality, most notably yielding worse FVD than vanilla diffusion. These findings align with our discussion in Section 3.2, showing that parameter sharing between condition understanding and target denoising imposes an unavoidable dual-role constraint, thereby further validating the necessity of our joint decoupling design.

Table 10: Necessity of architectural decoupling. Results comparing (1) Vanilla Diffusion, (2) Vanilla Diffusion with *training-only decoupling* (pretraining at $t = 1$ followed by fine-tuning across timesteps), and (3) ForeDiff with *both training- and architectural decoupling*.

| Method | FVD ↓ | PSNR ↑ | SSIM ↑ | LPIPS ↓ | STD$_{PSNR}$ ↓ | STD$_{SSIM}$ ↓ | STD$_{LPIPS}$ ↓ |
|---|---|---|---|---|---|---|---|
| Vanilla Diffusion | 53.8 | 27.1 | 88.2 | 5.65 | 0.66 | 1.33 | 0.65 |
| + Pretraining at $t = 1$ | 58.2 | 27.3 | 88.5 | 5.53 | 0.51 | 0.98 | 0.48 |
| ForeDiff | **51.5** | **27.4** | **88.8** | **5.25** | **0.37** | **0.70** | **0.35** |

**Necessity of two-stage training.** We suppose that jointly training the full architecture with a composite loss but not two-stage will reintroduces the entanglement problem that ForeDiff is designed

to resolve. To validate this empirically, we implemented the joint training variant using $\mathcal{L}_{\text{joint}} = \mathcal{L}_{\text{denoise}} + \lambda\mathcal{L}_{\text{deter}}$, and trained models on RoboNet with $\lambda = 0.1$ and $\lambda = 1$. The results in Table 11 reveal a clear pattern. With $\lambda = 0.1$, joint training brings only mild gains, but both the overall fidelity and the reduction of sample variance remain noticeably weaker than ForeDiff. Increasing the weight to $\lambda = 1$ strengthens variance suppression but begins to harm generative quality, as reflected by the increased FVD—showing that the predictive and generative streams start to interfere with one another. In contrast, the full ForeDiff design consistently achieves the strongest performance across all metrics, indicating that only the two-stage training decoupling can stabilize predictive representations and support effective denoising.

Table 11: Necessity of two-stage training. While joint training can offer mild gains, it ultimately converges to a compromised middle ground. The two-stage decoupled design of ForeDiff remains the most effective and stable solution for leveraging deterministic foresight while preserving diffusion's generative strengths.

| Method | FVD ↓ | PSNR ↑ | SSIM ↑ | LPIPS ↓ | STD$_{\text{PSNR}}$ ↓ | STD$_{\text{SSIM}}$ ↓ | STD$_{\text{LPIPS}}$ ↓ |
|---|---|---|---|---|---|---|---|
| Vanilla Diffusion | 53.8 | 27.1 | 88.2 | 5.65 | 0.66 | 1.33 | 0.65 |
| ForeDiff-zero | 52.7 | 27.2 | 88.4 | 5.54 | 0.68 | 1.36 | 0.66 |
| + joint training ( $\lambda = 0.1$ ) | 51.6 | 27.3 | 88.6 | 5.40 | 0.66 | 1.29 | 0.63 |
| + joint training ( $\lambda = 1$ ) | 52.7 | 27.3 | 88.6 | 5.36 | 0.48 | 0.92 | 0.44 |
| ForeDiff | **51.5** | **27.4** | **88.8** | **5.25** | **0.37** | **0.70** | **0.35** |

**Sensitivity to predictor quality.** To assess how sensitive ForeDiff is to the quality of the predictive stream, we retrained the generative stream using predictive streams saved at 0.5M and 0.8M iterations (instead of the final 1.0M model). Table 12 reports the quantitative results on RoboNet. Interestingly, even 50% of the predictor's training already provides clear improvements over vanilla diffusion, and the 0.8M predictor performs even slightly better than our default 1.0M configuration. This suggests two things:

- ForeDiff does not place strict requirements on predictor quality. Even a not-fully-converged predictor produces sufficiently structured intermediate representations to yield substantial gains.

- The default 1.0M predictor may be mildly overfitted; more careful tuning could unlock additional gains. However, to minimize implementation complexity and ensure consistency across datasets, we deliberately adopt a single, unified configuration without fine-grained tuning.

Table 12: Sensitivity to predictor quality. ForeDiff's two-stage design yields consistent improvements and imposing no critical requirements on the predictive pretraining phase.

| Method | FVD ↓ | PSNR ↑ | SSIM ↑ | LPIPS ↓ | STD$_{\text{PSNR}}$ ↓ | STD$_{\text{SSIM}}$ ↓ | STD$_{\text{LPIPS}}$ ↓ |
|---|---|---|---|---|---|---|---|
| Vanilla Diffusion | 53.8 | 27.1 | 88.2 | 5.65 | 0.66 | 1.33 | 0.65 |
| ForeDiff-zero | 52.7 | 27.2 | 88.4 | 5.54 | 0.68 | 1.36 | 0.66 |
| ForeDiff (predictive stream at 0.5M) | 51.5 | 27.2 | 88.6 | 5.35 | 0.48 | 0.91 | 0.45 |
| ForeDiff (predictive stream at 0.8M) | **50.9** | 27.3 | **88.8** | 5.26 | 0.39 | 0.74 | 0.37 |
| ForeDiff (predictive stream at 1M) | 51.5 | **27.4** | **88.8** | **5.25** | **0.37** | **0.70** | **0.35** |

**Calibration-oriented evaluation.** To further verify ForeDiff's reduced variability corresponds to better probabilistic modeling, rather than just collapsing to a single mode, we computed calibration-oriented metrics including CRPS, NLL and coverage curves, as shown in Table 13 and Figure 9. We observe that ForeDiff achieves lower CRPS and NLL, indicating that the samples become both more consistent and better calibrated. In other words, ForeDiff's reduced variance is not due to mode collapse but to more accurate conditional alignment. For coverage, vanilla diffusion appears closer to the ideal $y = x$ line, but this largely arises from inflated uncertainty: when the conditional mean is biased, a larger variance (directly reflected by larger STD) artificially improves coverage. In contrast,

ForeDiff yields narrower yet better-centered predictive distributions, leading to lower CRPS/NLL and more faithful calibration.

Table 13: Calibration evaluation. ForeDiff achieves improvement in both consistency and sample/distribution similarity, confirming that ForeDiff enhances predictive reliability without collapsing to a single deterministic mode.

| Dataset | Method | CRPS ↓ | NLL ↓ |
|---------|--------|--------|-------|
| RoboNet | Vanilla Diffusion | 0.0175 | 2.58 |
|         | ForeDiff | **0.0173** | **2.30** |
| RT-1 | Vanilla Diffusion | 0.0157 | -0.96 |
|      | ForeDiff | **0.0128** | **-1.19** |

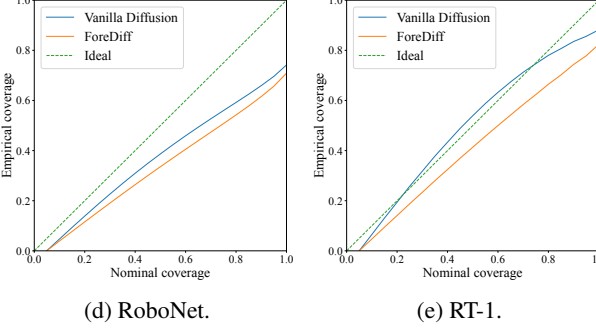

(d) RoboNet.     (e) RT-1.

Figure 9: Coverage calibration curves.

## E  LIMITATIONS AND FUTURE WORK

While Foresight Diffusion demonstrates improvements in both predictive accuracy and sampling consistency, several avenues remain open for further exploration and could inspire future work:

**Lack of large-scale validation.** We did not perform large-scale experiments involving substantially larger models or training datasets. This choice is primarily due to the substantial computational cost of scaling diffusion models, which often exceeds the capacity of academic research teams. Despite this, our setup follows established practices in prior work, and we believe the proposed method provides a fair and meaningful evaluation under moderate-scale settings. Importantly, our approach introduces algorithmic innovations that are orthogonal to scaling laws: ForeDiff does not alter the underlying generative paradigm, and thus its benefits should be complementary to improvements from scaling. A systematic study of scaling remains a valuable direction for future work.

**Focus on DiT-based architectures.** Our experiments focus on DiT-based diffusion backbones, which currently represent the state of the art in video modeling and offer a natural foundation for validating new ideas. Nevertheless, ForeDiff is not tied to DiT-specific components; its design—decoupling condition understanding from denoising—should in principle extend to alternative backbones such as CNN- or hybrid-based diffusion models. Exploring these directions may further broaden the applicability of our approach.

**Scope restricted to diffusion models.** This study centers on predictive diffusion models, which currently achieve state-of-the-art results on standard benchmarks. While alternative model families (e.g., auto-regressive or energy-based models) have demonstrated strengths in specific aspects, diffusion remains the most competitive framework in terms of overall predictive quality. Our work enhances diffusion further by improving both accuracy and consistency, ensuring it does not lag behind other approaches in robustness. We expect that the key insights of ForeDiff—particularly the disentanglement of condition understanding and target denoising—may generalize beyond diffusion, and extending similar architectural and training decoupling strategies to other generative paradigms is an exciting avenue for future research.

Taken together, these limitations highlight promising directions for scaling, architectural diversity, and model family generalization, which we hope will guide future advances in predictive modeling.

