# OpenReview forum: "Foresight Diffusion: Improving Sampling Consistency in Predictive Diffusion Models"
_ICLR.cc/2026/Conference — ICLR 2026 Poster_

### Official Review · Reviewer_Cbg4 · 2025-10-27

**Soundness:** 2
**Presentation:** 3
**Contribution:** 2
**Rating:** 4
**Confidence:** 4

**Summary:**

Diffusion models have achieved remarkable success in generative tasks, but they struggle with predictive learning, where sampling consistency with real trajectories is crucial. The authors identify this limitation as stemming from the entanglement of condition understanding and target denoising in shared architectures. To address this, they propose Foresight Diffusion (ForeDiff), which separates deterministic condition processing from stochastic denoising, leading to improved predictive accuracy and sampling consistency across robotic and spatiotemporal forecasting tasks.

**Strengths:**

* This paper is overall well-written and easy to follow. The authors start from the discussion of the difference between standard generation tasks and predictive tasks via diffusion model. The predictive tasks require more accuracy on the dynamic, while diffusion model’s generation process has a large uncertainty. The investigated problem is interesting and practical.

* The motivation of the proposed method, which is to detangle the generation and conditional guidance, is reasonable and clear.

* The empirical results demonstrate the good results over vanilla diffusion model baseline on predictive tasks.

**Weaknesses:**

* While the motivation is clear and conceptually sound, the qualitative justification remains weak. For instance, in Figure 3(b), the authors use LPIPS to measure the similarity between generated and ground-truth samples, concluding that diffusion models exhibit greater uncertainty as they produce both the best and worst cases compared to iVideoGPT. However, it is unclear whether LPIPS is an appropriate metric for assessing dynamic prediction quality. LPIPS primarily measures perceptual similarity, which may not capture temporal or dynamic consistency crucial for predictive tasks.

* The proposed methodological framework appears rather simple and incremental. The main difference lies in injecting the conditioning information into the first DiT block instead of concatenating it with the noised target input. Similar conditioning mechanisms have been used in many existing diffusion models (e.g., Stable Diffusion). If the primary novelty lies only in altering where the condition is injected, the technical contribution seems quite limited.

* The experimental evaluation also lacks breadth. The authors only compare their method against a vanilla diffusion baseline. Without comparisons to other relevant or stronger baselines, it is difficult to convincingly demonstrate the effectiveness of the proposed approach.

**Questions:**

1. What is the difference between the conditioning injection mechanism in Stable Diffusion and that in ForeDiff-Zero? If they are similar, does Stable Diffusion already possess the capability to disentangle generation and conditional guidance?

2. How does the proposed method compare to using a deterministic predictor directly for prediction tasks?

3. Many text-to-diffusion methods also use textual information as conditional input. How would textual conditioning be incorporated into ForeDiff-Zero?

4. The predictive task discussed here resembles the concept of a world model, which has recently gained significant attention. What is the conceptual and practical difference between your method and existing world models?

---

> ### Author Response · Authors · 2025-11-19
> **Response to Reviewer Cbg4 (Part I)**
>
> We sincerely thank Reviewer Cbg4 for providing insightful reviews and valuable comments. Below, we provide detailed responses to your questions.
>
> **[W1] On LPIPS and the qualitative illustration in Figure 3(b)**
>
> We agree that LPIPS is not a metric specifically designed for evaluating temporal or dynamic consistency. To point out, our use of LPIPS (together with PSNR/SSIM/FVD) in Fig. 3(b) follows the **standard practice in video prediction and predictive diffusion literature** for measuring reconstruction fidelity and perceptual similarity, which is **widely adopted in previous works** [1-4].
>
> To complement this, we also computed a simple temporal-consistency indicator: **the mean squared error between consecutive-frame differences $\mathcal{L} _ {\text{temporal-consistency}}=\sum _ t ||(x _ t-x _ {t-1})-(\hat{x} _ t-\hat{x} _ {t-1}) || _ 2$**.
>
> - Vanilla diffusion: 3.20e-3
> - ForeDiff: **3.06e-3**
>
> Such results provide additional evidence that ForeDiff achieves **smoother and more temporally coherent predictions**, indicating the improvement of ForeDiff is comprehensive. This is natural, since ForeDiff—similar to [1–4]—is designed **not merely to optimize per-frame quality but** to improve the fidelity of **the entire predicted trajectory**.
>
> Furthermore, in Fig. 3(b), **rather than a metric itself**, the inconsistency lies in the **variation of a metric** (such as LPIPS) across generated samples. Diffusion models often produce both very close and very distant samples relative to ground truth, reflecting high sampling uncertainty. This trend holds for other metrics as well (see Table 1), and we adopt LPIPS here for its simplicity.
>
> Overall, LPIPS in Fig. 3(b) is used solely to illustrate **intra-sample variability**, whereas our full quantitative evaluation and additional temporal analysis support the claim that ForeDiff improves both predictive fidelity and consistency.
>
> [1] Gupta A, Tian S, Zhang Y, et al. Maskvit: Masked visual pre-training for video prediction, 2022.
>
> [2] Wu B, Nair S, Martin-Martin R, et al. Greedy hierarchical variational autoencoders for large-scale video prediction, 2021.
>
> [3] Babaeizadeh M, Saffar M T, Nair S, et al. Fitvid: Overfitting in pixel-level video prediction, 2021.
>
> [4] Wu J, Yin S, Feng N, et al. ivideogpt: Interactive videogpts are scalable world models, 2024.
>
>
>
>
> **[W2, Q1] Novelty beyond injection position; why Stable Diffusion is fundamentally different**
>
> We emphasize that ForeDiff’s contribution is not limited to relocating where conditioning is injected. The essential difference lies in the **purpose and nature of the two-stage design**. Although both approaches use pre-trained models, ForeDiff fundamentally differs from prior works in the aspects of **motivation and training scheme**:
>
> - Prior conditional diffusion models (e.g., Stable Diffusion) aim to achieve **cross-modal alignment, where a natural and fundemental idea** is to use **unified representation encoders**. These encoders are pretrained in a task-agnostic manner and serve to extract high-level semantics from heterogeneous modalities and tasks. In these models, no entanglement issue arises since the objectives of the pretraining and diffusion training are fundamentally different.
> - In contrast, ForeDiff targets enhancing diffusion models’ predictive ability within the same predictive task, and proposes a **prediction-aware supervised pretraining**, which is **far from common practices** in single-modal predictive learning. The deterministic predictive stream is trained solely to capture future dynamics instead of general representations for observations.
>
> Moreover, our experiments show that the decoupling—not the conditioning location alone—is the key factor. Simply adjusting the injection position (as in ForeDiff-Zero) yields only modest gains, whereas introducing both architectural and training decoupling leads to substantially improved sampling consistency and predictive accuracy. This confirms that ForeDiff introduces a **new structural disentanglement tailored for predictive learning**, rather than reusing conditioning mechanisms from cross-domain diffusion models.

---

> ### Author Response · Authors · 2025-11-19
> **Response to Reviewer Cbg4 (Part II)**
>
> **[W3] Breadth of baseline comparisons**
>
> We would like to clarify that our evaluation does not rely solely on vanilla diffusion. In fact, **Table 3 already includes several widely-used and competitive predictive baselines**, such as *MaskViT*, *FitVid*, and *iVideoGPT*, which represent state-of-the-art approaches across robotic video forecasting and spatiotemporal prediction. ForeDiff achieves **superior or comparable performance across all of these methods**, indicating that its advantages are not tied to diffusion-only comparisons.
>
> Our focus in the main narrative is on the diffusion baseline because it directly reveals the effect of architectural disentanglement within diffusion models—our core contribution. But in terms of breadth, the paper already includes comparisons against strong, diverse baselines spanning transformer-based and autoregressive predictive models. These results collectively demonstrate that ForeDiff provides consistent improvements over a broad range of established predictive approaches, not just vanilla diffusion.
>
>
>
> **[Q2] Comparison to a purely deterministic predictor**
>
> Purely deterministic predictors can achieve strong per-frame reconstruction, but they fundamentally **cannot model uncertainty or generate diverse plausible futures**, making them unsuitable for predictive modeling tasks where the goal is to learn a *distribution* over future trajectories rather than a single point estimate. As illustrated in **Figure 1**, deterministic predictors lie at one extreme of the spectrum—high accuracy but collapsed to a single outcome—while vanilla diffusion models lie at the other—high diversity but overly unstable predictions.
>
> ForeDiff is explicitly designed to **strike a balance between these extremes**. By introducing a deterministic predictive stream only as a *guidance* mechanism, while retaining a stochastic diffusion process for generation, ForeDiff preserves the expressive uncertainty modeling of diffusion models but suppresses the excessive variance that harms predictive consistency.
>
> We also empirically compare ForeDiff with a deterministic ViT predictor of comparable capacity. The results are summarized below:
>
> | | Robonet  | | | RT1      | | | HeterNS     |
> | -------- | -------- | ------------ | ------------ | -------- | ------------ | ------------ | ----------- |
> | | FVD      | LPIPS (Mean) | LPIPS (Best) | FVD      | LPIPS (Mean) | LPIPS (Best) | Relative L2 |
> | ForeDiff | **51.5** | 5.25         | 4.51         | **12.0** | 3.42 | 3.11         | **0.176**   |
> | Deterministic ViT | 83.2     | **5.20**         | 5.20         | 20.2     | **3.24** | 3.24         | 0.191       |
>
> Although the deterministic predictor shows competitive *mean* LPIPS—unsurprising because it collapses to an average—it performs **substantially worse on FVD**, indicating degraded temporal coherence and video realism. These results confirm that deterministic prediction alone cannot achieve the distributional fidelity required in predictive modeling.
>
> In summary, deterministic predictors lie at one side of the balance. ForeDiff combines deterministic guidance with stochastic generation, approaching deterministic-level predictive accuracy while retaining diffusion-level generative capability.

---

> ### Author Response · Authors · 2025-11-19
> **Response to Reviewer Cbg4 (Part III)**
>
> **[Q3] Incorporating textual conditioning into ForeDiff**
>
> Textual conditioning can be incorporated into ForeDiff **without any architectural modification**. Our method does not alter the underlying diffusion backbone or the conditioning interfaces—it only reorganizes how conditioning is used (deterministic foresight first, stochastic generation second). Therefore, any conditioning modality already supported by standard latent diffusion architectures naturally carries over.
>
> Concretely, textual input can be handled exactly the same way as in conventional text-to-image or text-to-video diffusion:
>
> 1. A text encoder (e.g., CLIP, T5) produces text embeddings.
> 2. These embeddings pass through the **predictive stream**, which acts as a deterministic conditioner.
> 3. The resulting intermediate features—**not modified in structure**—are then used as conditioning signals for the generative stream.
>
> No new modules, fusion layers, or cross-attention mechanisms need to be redesigned; the same conditioning pathways used in standard latent diffusion remain intact. ForeDiff simply changes *when* the conditioning representation is learned (pretrained deterministically) and *how* it is applied (frozen foresight features), not *what* the conditioning representation looks like.
>
> Finally, we emphasize that **ForeDiff is specifically designed for predictive tasks**, where sampling consistency and trajectory stability are essential. In contrast, text-to-image/video models typically desire diversity—as illustrated in **Figure 1**, generative tasks benefit from broad sample variation. For such use cases, the variance-reducing nature of ForeDiff may not be desirable. Thus, while textual conditioning is fully compatible with ForeDiff, its advantages are most relevant in **predictive dynamics modeling**, not open-ended creative generation.
>
>
>
> **[Q4] Relation to world models**
>
> World models describe a **task setting**—modeling/predicting future environment states—rather than prescribing a particular architecture. ForeDiff, by contrast, is a **modeling and training paradigm** that can serve as the base predictor for world-model tasks, but is not limited to them. In our experiments, ForeDiff is applied not only to robotic video prediction (a common world-model scenario) but also to scientific spatiotemporal forecasting, which lies outside typical world-model formulations.
>
> Several baselines in **Table 3** (e.g., iVideoGPT) originate from world-model research, and ForeDiff consistently outperforms or matches them without incorporating world-model–specific components. Thus, ForeDiff could be viewed as a general predictive modeling approach that can instantiate a world model when applied to such tasks.

---

### Official Review · Reviewer_SCh5 · 2025-10-30

**Soundness:** 2
**Presentation:** 2
**Contribution:** 2
**Rating:** 2
**Confidence:** 3

**Summary:**

The paper proposes Foresight Diffusion, which decouples a deterministic predictive stream (ViT blocks processing the condition $y$) from a generative/denoising stream (DiT blocks processing $x_t$), and then adopts a two-stage training scheme: first train the predictive stream as a standalone deterministic predictor with a prediction head, then freeze it and feed its internal representation $g_M$ to guide diffusion generation. The claim is that this design improves “predictive ability” and sampling consistency (lower across-seed variance of PSNR/SSIM/LPIPS) on datasets such as RoboNet, RT-1, and HeterNS.

**Strengths:**

The method is simple and relatively easy to reproduce: the architectural split and the two-stage schedule are clearly described, and the paper includes ablations on the prediction head and the number of ViT blocks in the predictive stream. Some datasets show moderate improvements in PSNR/LPIPS and reduced reported variability, and removing the prediction head appears beneficial across tasks per the appendix tables.

**Weaknesses:**

(1) Limited novelty and weak theory. The main idea—strengthening a condition encoder via standalone pretraining and then freezing its features to condition the denoiser—tracks common practice in conditional diffusion and teacher-feature guidance. Theoretical support is thin: the central formal argument reduces the $t=1$ case to a deterministic model by zeroing the first-layer weights, which does not yield general consistency or error bounds for multi-step diffusion.

(2) Consistency is proxied narrowly and may conflate “agreement” with “accuracy.” Using the STD of PSNR/SSIM/LPIPS across repeated samples risks rewarding collapse or overly strong reliance on the condition signal. The paper lacks calibration-oriented metrics (e.g., NLL/CRPS/coverage) or uncertainty diagnostics to demonstrate that reduced variability corresponds to better probabilistic modeling, rather than just more concentrated—but possibly biased—outputs.

(3) Distributional quality gains are not robust. On RT-1, the reported FVD degrades as predictive blocks increase (e.g., baseline 11.7 vs. 12.0 at the default $M=6$, undermining the claim that decoupling and two-stage training consistently improve generative quality. The paper should transparently analyze this trade-off.

(4) Ablations do not fully isolate training-scheme and compute confounds. Although the appendix contrasts “training-only decoupling” (pretrain at $t=1$ then fine-tune) with the architectural variant, the study still does not establish equalized budgets (steps/FLOPs/memory/denoising steps) between end-to-end shared vs. decoupled + frozen designs. Nor are throughput/latency or memory overheads reported for the extra predictive stream and the freeze-and-feed pipeline.

(5) Baselines and task breadth are insufficient. The paper mostly compares to its own variants and standard conditional diffusion. For scientific/physical forecasting, head-to-head evaluations against strong PDE/forecasting models (e.g., modern operator-learning and transformer baselines) are missing; for video prediction, broader masked/autoregressive competitors would better position the claimed advantages.

(6) Method details and limits remain under-explained. The Fusion mechanism and temporal conditioning choices are specified but not systematically stress-tested for long-horizon prediction or out-of-distribution conditions; appendix tables list numbers (e.g., PredHead effects) without enough analysis about when and why the components help or hurt.

**Questions:**

Can you report calibration metrics (e.g., NLL/CRPS/coverage) and analyze their relationship to the STD of PSNR/SSIM/LPIPS, on the same hardware and sampling budgets?

Please provide equal-budget comparisons (same steps, FLOPs, memory, sampling steps) between (i) end-to-end shared models and (ii) your decoupled two-stage pipeline, along with throughput/latency and memory use at training and inference.

---

> ### Author Response · Authors · 2025-11-19
> **Response to Reviewer SCh5 (Part I)**
>
> We sincerely thank Reviewer SCh5 for providing insightful reviews and valuable comments. Below, we provide detailed responses to your questions.
>
> **[W1] Novelty upon common practice and soundness of theory**
>
>
> We thank the reviewer for the insightful assessment. However, we believe there may be some misunderstanding regarding the novelty and theoretical contributions of our work.
>
> The idea of *incorporating a condition encoder* in ForeDiff is **practically simple but novel, non-trivial relative to common practice**, especially in the context of predictive diffusion models. Although both approaches use pre-trained models, ForeDiff fundamentally differs from prior works in the aspects of **motivation and training scheme**:
> - Prior conditional diffusion models (e.g., Stable Diffusion) aim to achieve **cross-modal alignment, where a natural and fundemental idea** is to use **unified representation encoders**. These encoders are pretrained in a task-agnostic manner and serve to extract high-level semantics from heterogeneous modalities and tasks. In these models, no entanglement issue arises since the objectives of the pretraining and diffusion training are fundamentally different.
> - In contrast, ForeDiff targets enhancing diffusion models’ predictive ability within the same predictive task, and proposes a **prediction-aware supervised pretraining**, which is **far from common practices** in single-modal predictive learning. The deterministic predictive stream is trained solely to capture future dynamics instead of general representations for observations.
>
> Overall, our work is not following *common practice* but **proposing a novel structural disentanglement tailored for predictive diffusion**. This addresses a specific problem in standard diffusion models—entangling condition understanding and target denoising—that has not been analyzed before. If similar approaches have been proposed in diffusion-based prediction literature, we would appreciate references.
>
> Further, we would like to clarify that the theoretical analysis in Lemma 3.1 is **not intended to be a formal bound, but rather an interpretive observation**. It provides a straightforward and intuitive explanation for the behavior of diffusion models in predictive tasks, and we position this as an explanatory result that complements the empirical findings. Though analyzing *general consistency or error bounds* would also be helpful, it is far outside the scope of this work that **focuses on specific predictive tasks and empirical demonstrations**.
>
>
>
> **[W2, Q1] Conflict between *agreement* and *accuracy*; need for calibration metrics**
>
> We would like to clarify that *agreement* and *accuracy* are not conflicting objectives in our setting. In fact, ForeDiff **improves both** simultaneously: the reduction of variance across samples (lower STD of PSNR/SSIM/LPIPS) reflects stronger agreement, while the consistent improvement in average performance metrics (e.g., higher PSNR, lower LPIPS) demonstrates higher accuracy. Here, the use of LPIPS/PSNR/SSIM/FVD follows strictly from prior work in video prediction to measure sample quality, and the variance of these metrics further reflects sampling consistency.
>
> Nonetheless, we acknowledge the reviewer’s request for *calibration-oriented evaluation*, and we therefore computed **CRPS**, **NLL**, and **coverage** metrics (reported in the revised paper, page 20):
>
> |                   | RoboNet    |          | RT-1       |           |
> | ----------------- | ---------- | -------- | ---------- | --------- |
> |                   | CRPS       | NLL      | CRPS       | NLL       |
> | Vanilla Diffusion | 0.0175     | 2.58     | 0.0157     | -0.96     |
> | ForeDiff          | **0.0173** | **2.30** | **0.0128** | **-1.19** |
>
> We observe that ForeDiff achieves lower CRPS and NLL, indicating that the samples become both **more consistent and better calibrated**. In other words, ForeDiff’s reduced variance is not due to mode collapse but to more accurate conditional alignment. For coverage, vanilla diffusion appears closer to the ideal $y=x$ line, but this largely arises from **inflated uncertainty**: when the conditional mean is biased, a larger variance (directly reflected by larger STD) artificially improves coverage. In contrast, ForeDiff yields narrower yet **better-centered** predictive distributions, leading to lower CRPS/NLL and more faithful calibration.
>
> Overall, ForeDiff achieves improvement in both consistency (*agreement*) and sample/distribution similarity (*accuracy*), confirming that ForeDiff enhances predictive reliability without collapsing to a single deterministic mode.

---

> ### Author Response · Authors · 2025-11-19
> **Response to Reviewer SCh5 (Part II)**
>
> **[W3] Distributional quality degradation and predictive–generative trade-off**
>
> We acknowledge the observation that the FVD metric slightly degrades on RT-1. This behavior is expected and represents a **controlled trade-off** between *predictive consistency* and *generative diversity*. FVD measures the overall realism and temporal coherence of generated videos, emphasizing **generative ability**, whereas metrics such as LPIPS or PSNR primarily capture **predictive accuracy** and per-frame fidelity. Our goal in ForeDiff is to decouple these two aspects—prediction and generation—so that each can be learned more effectively.
>
> To illustrate this trade-off, we here compare ForeDiff with a deterministic ViT predictor of identical scale:
>
> | | RoboNet  | | | RT-1     | | |
> | ------- | -------- | ------ | ----- | --- | -- | --- |
> | | FVD | LPIPS (Mean) | LPIPS (Best) | FVD | LPIPS (Mean) | LPIPS (Best) |
> | ForeDiff | **51.5** | 5.25 | 4.51 | **12.0** | 3.42 | 3.11 |
> | Deterministic ViT | 83.2 | **5.20** | 5.20 | 20.2 | **3.24** | 3.24 |
>
> The deterministic predictor achieves slightly better average LPIPS, as it collapses to a single trajectory and thus eliminates sample variance. However, its FVD scores are dramatically worse, reflecting a loss of temporal realism and generative richness. In contrast, ForeDiff maintains competitive FVD while achieving much lower variance and stronger predictive fidelity.
>
> As illustrated in **Figure 1**, ForeDiff deliberately **strikes a balance** between two extremes: (i) deterministic predictors with strong accuracy but no generative diversity, and (ii) vanilla diffusion models with high diversity but suboptimal predictive coherence. The mild FVD change is therefore an *intentional trade-off*, showing that ForeDiff approaches deterministic-level predictive accuracy while retaining diffusion-level generative capability. For predictive modeling tasks such as robotic video forecasting or scientific simulation—where stable and consistent trajectories matter more than unrestricted diversity—this trade-off represents a **desirable optimization direction** rather than a limitation.
>
> Importantly, our additional experiments on RoboNet (see response to Reviewer bbPz, [W1, Q1]; or revised paper, page 19) show that using an earlier predictive-stream checkpoint (e.g., 0.8M iterations) actually yields **better FVD** than the default 1.0M setting. This demonstrates that **ForeDiff has headroom for further improving distributional quality**, and the RT-1 behavior should not be interpreted as a structural limitation of the method. For the main paper, we intentionally keep a **single unified configuration across datasets** to ensure fairness and reproducibility rather than tuning per-dataset performance.
>
>
>
> **[W4, Q2] Compute fairness and efficiency**
>
> We would like to clarify that, though our ForeDiff introduces an extra predictive stream and additional parameters (≈+50%, regardless of the pretrained autoencoder), the budgets in ForeDiff and vanilla diffusion baselines in our experiments are kept comparable in two ways:
>
> 1. **Training: no repeated computation in stage-2.** Once the predictive stream is obtained, we perform **only a single forward pass to cache** its outputs for the training set. The predictive stream thus is not re-evaluated at every iteration during the generative training. The only extra training cost comes from stage-1 predictive training, which is substantially smaller than the end-to-end cost of many larger baselines. This ensures the overall training cost similar to the vanilla diffusion baseline. Moreover, the FVD-params scatter in **Fig 3(a)** shows that our vanilla diffusion baseline already lies on the **efficiency frontier** (low params / low FVD); ForeDiff improves accuracy and sampling consistency on top of this efficient point, making the modest extra stage-1 cost acceptable for predictive tasks.
> 2. **Inference: negligible inference overhead.** At test time, the predictive stream runs **only once** per sequence, whereas the generative stream iterates **50 denoising steps**. The extra runtime due to the predictive pass is therefore ≈ **1%** of total sampling time, while memory usage remains unchanged during the iterative denoising phase.
>
> To further ensure that the improvements stem from **architectural design rather than parameter scaling**, we followed the protocol in **Section 4.3**. Specifically, we extended the vanilla diffusion model to match ForeDiff’s layer count (18 DiT blocks) and compared it to ForeDiff. As shown in **Figure 8(c)**, ForeDiff outperforms the scaled-up vanilla diffusion by a substantial margin, confirming that the gains originate from the **synergy between deterministic prediction and conditional diffusion**, not from model size alone.
>
> Overall, the predictive stream incurs a modest, **one-time training cost and negligible inference overhead**, and it delivers lasting and worthwhile benefits.

---

> ### Author Response · Authors · 2025-11-19
> **Response to Reviewer SCh5 (Part III)**
>
> **[W5] Baseline coverage and task breadth**
>
> For **video prediction**, we have already provided comprehensive comparisons in **Table 3**, which includes most of the strong baselines commonly used in this domain (e.g., *MaskViT*, *FitVid*, *iVideoGPT*). Across all metrics, ForeDiff demonstrates superior or comparable performance, confirming the robustness of its predictive improvements.
>
> For **scientific or physical forecasting**, we note that direct comparison with classical scientific models would be **misleading** since (i) there is a **lack of evaluation for large-scale** scientific models and (ii) some models (such as PINN) often rely on explicit physical equations, which is **fundamentally different** from our data-driven prediction setup. For instance, comparing to UniSolver [1] (a PDE-Conditional Transformer) which achieves a relative L2 of 0.0263 on HeterNS, even our vanilla diffusion is undoubtedly better than it. Therefore, we believe it would be more beneficial to compare with diffusion-based baselines that share similar modeling assumptions.
>
> Our evaluation scope therefore focuses on **diffusion-based predictive models**, within which compute budgets, architecture families, and objectives are directly comparable, and can directly reveal the effect of architectural disentanglement within diffusion models—our core contribution. The consistent improvements across RoboNet, RT-1, and HeterNS demonstrate that ForeDiff’s design generalizes across diverse video and physical prediction settings, while keeping compute and modeling assumptions **fair and well-controlled**.
>
> [1] Zhou H, Ma Y, Wu H, et al. Unisolver: PDE-conditional transformers are universal PDE solvers, 2025.
>
>
>
> **[W6] Method details and design clarification**
>
> We believe the under-explaining concern arises from a misunderstanding of where our analysis is presented. The discussions related to the appendix tables (e.g., the effect of the PredHead module) are covered in the main text, **Section 4.3 Analysis**, not only in the appendix. In particular, we examine whether ForeDiff benefits more from the **predictive ability of the ViT stream** or from its **explicit prediction outputs**. As shown in **Figure 8(a)**, conditioning the DiT stream on the PredHead outputs leads to a clear performance drop, confirming that the learned internal representations—not the regression outputs—provide stronger guidance for the generative process.
>
> Regarding the *fusion mechanism and temporal conditioning*, both are intentionally designed to be lightweight and stable rather than heavily tuned. The using of MLP in fusion is natural and the temporal condioning mechanism follows common practice in video diffusion models [2]. Moreover, as ForeDiff is proposed as an **architectural and training-scheme improvement**, the block-level implementation choices are not the primary focus of our study, and we therefore keep them minimal and standardized to ensure fair comparisons. While we did not include extensive long-horizon or out-of-distribution stress tests, ForeDiff was evaluated across three domains—RoboNet, RT-1, and HeterNS—which differ substantially in dynamics, stochasticity, and temporal scales. The consistent performance trends across these datasets suggest that our architectural design generalizes beyond specific task configurations.
>
> [2] Blattmann A, Dockhorn T, Kulal S, et al. Stable video diffusion: Scaling latent video diffusion models to large datasets, 2023.

---

### Official Review · Reviewer_6hoQ · 2025-10-31

**Soundness:** 3
**Presentation:** 4
**Contribution:** 3
**Rating:** 8
**Confidence:** 3

**Summary:**

This paper introduces Foresight Diffusion (ForeDiff), a framework designed to improve sampling consistency in predictive diffusion models. The authors identify that vanilla diffusion models, while effective for diverse generation tasks, suffer from high sample variance when applied to predictive learning where consistency is crucial. ForeDiff addresses this by architecturally decoupling condition understanding from target denoising through separate predictive and generative streams, and employing a two-stage training scheme with deterministic pretraining.

**Strengths:**

- The paper presents a novel architectural approach by explicitly separating condition processing from denoising, which is a creative departure from standard conditional diffusion models.
- The focus on sampling consistency as a distinct requirement for predictive tasks versus generative tasks is an important problem formulation.
- Clear mathematical formulation and proof of the key lemma connecting diffusion and deterministic models
- Comprehensive experimental evaluation across three diverse datasets (RoboNet, RT-1, HeterNS) covering both real-world robotics and scientific computing domains
- Shows consistent (but modest) improvements across multiple metrics and datasets
- Well-structured paper with clear motivation through visual examples (Figures 1 and 2) and illustration of architectural differences (Figure 4)

**Weaknesses:**

- As acknowledged by the authors, experiments are limited to moderate-scale settings (64×64 resolution, relatively small models).
- Only DiT-based architectures are evaluated; generalization to U-Net or other diffusion backbones is assumed but not demonstrated
- The connection between predictive ability and sampling consistency could be more rigorously established
- The improvement margins, while consistent, are sometimes modest
- The two-stage training could be seen as unfair comparison since vanilla diffusion does not benefit from deterministic pretraining

**Questions:**

- Do you anticipate ForeDiff will scale well to higher-resolution datasets and larger models (e.g., 256×256 or video generation)?
- Have you attempted to adapt ForeDiff to non-DiT backbones, such as U-Net or latent diffusion architectures? If not, what are the main obstacles?
- Since ForeDiff benefits from deterministic pretraining, could you provide a control where the baseline diffusion model is pretrained with a similar deterministic phase?
- Could you clarify why the deterministic predictor must be trained separately rather than jointly fine-tuned with the diffusion model?

---

> ### Author Response · Authors · 2025-11-19
> **Response to Reviewer 6hoQ (Part I)**
>
> We sincerely thank Reviewer 6hoQ for providing insightful reviews and valuable comments. Below, we provide detailed responses to your questions.
>
> **[W1, Q1] Scaling to larger resolutions and larger models**
>
> ForeDiff’s architectural mechanism—decoupling predictive conditioning from generative denoising—is **inherently scale-agnostic**. It does not rely on any 64×64–specific design choices, and the way predictive features guide the diffusion process operates identically at higher resolutions and larger model capacities. Therefore, we expect ForeDiff to transfer naturally to higher-resolution predictive tasks such as 128×128 or 256×256.
>
> We are currently training **RT-1 at 128×128**, with results expected within several days. The preliminary learning curves exhibit the same trend observed at 64×64, and we will report the results once they are finalized.
>
> We acknowledge that large-scale validation (e.g., 256×256 with substantially larger models) is limited by computational cost—scaling diffusion models from 64×64 to 128×128 increases token length by 4× and total compute by roughly **16×**. This constraint is common across predictive diffusion research and is discussed in **Appendix E**. Crucially, our contribution is **algorithmic rather than scale-dependent**: the architectural disentanglement in ForeDiff is orthogonal to model size and should remain beneficial as models and resolutions scale up.
>
>
>
> **[W2, Q2] Generalization to non-DiT backbones**
>
> ForeDiff is **not inherently tied to DiT**. The core idea—decoupling a deterministic predictive stream from a generative diffusion stream—applies equally to architectures such as U-Nets as long as intermediate features can be used as conditioning signals. Nothing in our design relies on DiT-specific components. In fact, our implementation already follows the **latent diffusion paradigm**, where both the predictive and generative streams operate in the latent space rather than pixels.
>
> In this work, we focus on DiT because it is **the dominant backbone in recent predictive diffusion research** and provides a clean, transformer-based structure for controlled comparisons. Using a single family of architectures keeps the study focused on ForeDiff’s contribution rather than on backbone differences. Adapting the method to other backbones is mainly an engineering effort rather than a conceptual challenge, and we leave such extensions to future work given resource constraints.
>
>
>
> **[W3] Connection between predictive ability and sampling consistency**
>
> Our view is simple: **a more accurate predictive signal naturally reduces the uncertainty that the diffusion process must resolve**, leading to tighter, more stable samples. As discussed in Lemma 3.1 and Section 3.2, standard diffusion models entangle condition understanding with denoising across all timesteps, which weakens the model’s effective predictive ability and increases sample variance. By introducing a dedicated deterministic predictive stream, ForeDiff provides a **sharper and more disentangled conditional representation**, which the diffusion stream can sample around more reliably. Empirically, this connection is reflected consistently: ForeDiff improves both mean accuracy (PSNR/LPIPS) and cross-sample variance, and calibration metrics (CRPS/NLL) further show that the reduced variance corresponds to **better-centered predictive distributions**, not mode collapse.
>
> In short, enhancing predictive ability through disentanglement directly improves sampling consistency, and our results across three domains support this relationship.
>
>
>
> **[W4] Magnitude of improvements**
>
> We would like to clarify that the metric improvement over vanilla diffusion model baselines is not merely a marginal gain but a substantial one. *In view of statistical significance*, a paired t-test on avg LPIPS scores in Figure 2 have demonstrated that ForeDiff achieves a **t-value of 9.0675** compared to vanilla diffusion, which indicates significantly better performance. *In view of practical significance*, we observe that such improvement achieves similar magnitude as those between various baseline methods in Table 3, where recently proposed methods **also** achieve an improvement on the order of 0.1 in PSNR, SSIM and LPIPS. These suggest that ForeDiff’s advantage is not just a small tweak but a meaningful enhancement.

---

> ### Author Response · Authors · 2025-11-19
> **Response to Reviewer 6hoQ (Part II)**
>
> **[W5, Q3] Fairness of two-stage training & deterministic-pretraining control**
>
> We agree that a natural control is to give the vanilla diffusion baseline the **same deterministic pretraining advantage**. To address this, we performed the experiment reported in **Appendix E (Table 10)** in our original manuscript: we first pretrained a standard diffusion model **only at timestep t=1**, allowing it to behave as a deterministic predictor (per Lemma 3.1), and then fine-tuned it across all timesteps. This setup mirrors ForeDiff’s training-scheme decoupling *without* architectural separation.
>
> This control indeed improves sampling consistency (lower variance), yet it **still underperforms** ForeDiff and even shows **worse FVD than vanilla diffusion**. These results show that **training-scheme decoupling alone is insufficient**: the predictive and generative roles remain entangled in a single network, preventing the formation of stable foresight representations. Only **architectural + training decoupling together** yield the benefits observed in ForeDiff.
>
>
>
> **[Q4] Why the predictor must be trained separately rather than jointly fine-tuned**
>
> Joint optimization **reintroduces the entanglement** problem that ForeDiff is designed to resolve. When the predictive and generative objectives are optimized simultaneously, the potential learning conflicts would result in a suboptimal information extraction from the predictive stream. This coupling counteracts the core motivation behind our two-stage scheme.
>
> To validate this empirically, we implemented the joint training variant using $\mathcal{L} _ {\text{joint}} = \mathcal{L} _ {\text{denoise}} + \lambda \mathcal{L} _ {\text{deter}}$, and trained models on RoboNet with λ=0.1 and λ=1.
>
> |                            | FVD      | PSNR     | SSIM     | LPIPS    | $\text{STD}_\text{PSNR}$ | $\text{STD}_\text{SSIM}$ | $\text{STD}_\text{LPIPS}$ |
> | ------- | -------- | -------- | -------- | -------- | ------- | -------- | ----- |
> | Vanilla Diffusion | 53.8     | 27.1     | 88.2     | 5.65     | 0.66 | 1.33 | 0.65 |
> | ForeDiff-zero | 52.7     | 27.2     | 88.4     | 5.54     | 0.68 | 1.36 | 0.66 |
> | + joint training ( λ=0.1 ) | 51.6     | 27.3     | 88.6     | 5.40     | 0.66 | 1.29 | 0.63 |
> | + joint training ( λ=1 )   | 52.7     | 27.3     | 88.6     | 5.36     | 0.48 | 0.92 | 0.44 |
> | ForeDiff | **51.5** | **27.4** | **88.8** | **5.25** | **0.37** | **0.70** | **0.35** |
>
> These results reveal a clear pattern. With λ = 0.1, joint training brings only mild gains, but both the overall fidelity and the reduction of sample variance remain noticeably weaker than ForeDiff. Increasing the weight to λ = 1 strengthens variance suppression but begins to harm generative quality, as reflected by the increased FVD—showing that the predictive and generative streams start to interfere with one another. In contrast, the full ForeDiff design consistently achieves the strongest performance across all metrics, indicating that only the two-stage architectural and training decoupling can simultaneously stabilize predictive representations and support effective denoising.
>
> Overall, the experiments confirm that while joint training can offer mild gains, it ultimately converges to a **compromised middle ground**. The two-stage decoupled design of ForeDiff remains the most effective and stable solution for leveraging deterministic foresight while preserving diffusion’s generative strengths.

---

> ### Author Response · Authors · 2025-11-24
> **Response to Reviewer 6hoQ after Further Experiments**
>
> **Update: 128×128 RT-1 Results**
>
> We have now completed the additional experiments on **RT-1 at 128×128 resolution**. ForeDiff continues to outperform the vanilla diffusion model under this higher-resolution setting:
>
> |                   | FVD      | PSNR     | SSIM     | LPIPS    | $\text{STD}_\text{PSNR}$ | $\text{STD}_\text{SSIM}$ | $\text{STD}_\text{LPIPS}$ |
> | ----------------- | -------- | -------- | -------- | -------- | ------------------------ | ------------------------ | ------------------------- |
> | Vanilla Diffusion | 0.42     | 28.7     | 91.9     | 5.00     | 1.18                     | 1.69                     | 1.17                      |
> | ForeDiff          | **0.39** | **29.7** | **93.4** | **4.01** | **0.78**                 | **0.85**                 | **0.58**                  |
>
> These results are consistent with our observations at 64×64 resolution and further confirm that ForeDiff’s improvements arise from its **architectural design rather than scale-specific factors**.

---

### Official Review · Reviewer_bbPz · 2025-11-01

**Soundness:** 3
**Presentation:** 3
**Contribution:** 3
**Rating:** 4
**Confidence:** 3

**Summary:**

This paper addresses a key discrepancy between the typical use of diffusion models for generative tasks and their application to predictive learning. While generative tasks often value sample diversity, predictive tasks (like video forecasting) require sampling consistency, where generated samples are tightly clustered around a single, physically plausible ground-truth future. The authors hypothesize that the suboptimal consistency of standard predictive diffusion models stems from the entanglement of condition understanding and target denoising within shared architectures and co-training schemes.

To solve this, the paper proposes Foresight Diffusion (ForeDiff), a framework that disentangles these two roles. ForeDiff consists of two main contributions:

1. A Decoupled Architecture consists of a condition understanding branch and a generative branch.

2. A Two-Stage Training Scheme that enables the predictive branch develops a strong foresight which can later provide a strong conditioning signal to the generative branch.

Experiments on robotic video prediction and scientific spatiotemporal forecasting demonstrate that ForeDiff significantly improves both predictive accuracy and sampling consistency.

**Strengths:**

This is a clear and well-motivated paper. It identifies an important issue with applying diffusion models to predictive tasks and proposes an effective solution. The experimental results on various tasks and multiple ablations studies provide evidence for the authors' claims. This work can potentially be applied to various fields such as forecasting, robotics, and scientific ML.

**Weaknesses:**

1. The proposed two-stage training scheme, while effective, introduces additional complexity to the training pipeline compared to a single-stage, end-to-end model. It requires a separate pretraining phase for the predictive branch, which may add to the overall engineering effort, training time, and need for hyperparameter tuning. A discussion of this trade-off (implementation complexity vs. consistency gain) would be beneficial.

2. Using a frozen, deterministic predictor could also be a potential limitation. Predictive learning is not always purely deterministic; there can be legitimate stochasticity or multi-modality in the dataset. Forcing the model to foresee the future and training with a simple $L_2$ loss might suppress the model's ability to capture valid multi-modality, collapsing all potential futures into some form of averaged future prediction.

3. The generative network is fundamentally conditioned on the output of the frozen predictive stream. If the predictive stream makes a significant error, the generative network has no mechanism to correct this. It is forced to denoise towards a faulty premise. Are there any ways to mitigate such scenarios?

**Questions:**

1. Following on Weakness 1, how sensitive is the final ForeDiff model to the quality of the pretrained deterministic predictor from Stage 1? For instance, if the predictive branch is overfit, or trained for too few steps, does this significantly harm the generative stream's performance or consistency?

2. Could the authors comment on the potential of the setting of jointly training the full architecture with a composite loss, i.e.,
$\\mathcal{L}$ = $\mathcal{L}\_{denoise}$ + $\lambda \mathcal{L}\_{deter}$? This ablation studies can further demonstrate the effectiveness of the two-stage training method.

---

> ### Author Response · Authors · 2025-11-19
> **Response to Reviewer bbPz (Part I)**
>
> We sincerely thank Reviewer bbPz for providing insightful reviews and valuable comments. Below, we provide detailed responses to your questions.
>
> **[W1, Q1] Implementation complexity and sensitivity to predictor quality**
>
> We acknowledge that ForeDiff introduces an additional pretraining phase for the predictive stream, which modestly increases training complexity compared to a single-stage model. However, this stage uses the same data, optimizer, and loss function as the main task, **without any task-specific hyperparameter tuning or additional supervision**, making it easy to accomplish. All results were obtained in a **single training pass**, without separate tuning for the predictive and generative stages.
>
> In practice, the added stage imposes limited engineering overhead. Once trained, the predictive stream’s intermediate features are cached and reused during diffusion training, avoiding redundant computation. The inference-time overhead is negligible since the predictive stream runs only once while the diffusion process iterates for 50 denoising steps.
>
> To assess how sensitive ForeDiff is to the quality of the predictive stream, we retrained the generative stream using predictive streams saved at 0.5M and 0.8M iterations (instead of the final 1.0M model):
>
> |                                      | FVD      | PSNR     | SSIM     | LPIPS    | $\text{STD}_\text{PSNR}$ | $\text{STD}_\text{SSIM}$ | $\text{STD}_\text{LPIPS}$ |
> | --------| -------- | -------- | -------- | -------- | --------- | ---------- | ------ |
> | Vanilla Diffusion  | 53.8     | 27.1     | 88.2     | 5.65     | 0.66 | 1.33   | 0.65   |
> | ForeDiff-zero | 52.7     | 27.2     | 88.4     | 5.54     | 0.68 | 1.36 | 0.66 |
> | ForeDiff (predictive stream at 0.5M) | 51.5     | 27.2     | 88.6     | 5.35     | 0.48 | 0.91 | 0.45 |
> | ForeDiff (predictive stream at 0.8M) | **50.9** | 27.3     | **88.8** | 5.26     | 0.39 | 0.74 | 0.37 |
> | ForeDiff (predictive stream at 1M)   | 51.5     | **27.4** | **88.8** | **5.25** | **0.37** | **0.70** | **0.35** |
>
> Interestingly, even **50%** of the predictor’s training already provides clear improvements over vanilla diffusion, and the **0.8M** predictor performs *even slightly better* than our default 1.0M configuration. This suggests two things:
>
> 1. **ForeDiff does not place strict requirements on predictor quality.** Even a not-fully-converged predictor produces sufficiently structured intermediate representations to yield substantial gains.
> 2. The default 1.0M predictor may be mildly overfitted; more careful tuning **could potentially unlock additional gains**. However, to minimize implementation complexity and ensure consistency across datasets, we deliberately adopt a single, unified configuration without fine-grained tuning.
>
> Overall, ForeDiff’s two-stage design introduces minimal overhead while yielding consistent improvements and imposing no critical requirements on the predictive pretraining phase.
>
>
>
> **[W2] Deterministic predictor and *multi-modality***
>
> We agree that predictive learning problems may contain legitimate stochasticity or multi-modality. The deterministic predictive stream in ForeDiff is **not intended to eliminate such variability**, but to provide the diffusion model with a **more focused and coherent prior**, preventing the generative process from drifting into implausible futures. Importantly, the generative stream remains fully probabilistic and does not regress to the predictor’s deterministic output. It receives only the predictor’s **intermediate feature representations**, which encode coarse future dynamics and spatial-temporal structure. These features act as a soft, high-level guidance signal, while the diffusion process still samples stochastic refinements over the denoising steps. Thus, the predictor constrains the solution space to physically plausible futures without suppressing legitimate multi-modality.
>
> |                   | RoboNet    |          | RT-1       |           |
> | ----------------- | ---------- | -------- | ---------- | --------- |
> |                   | CRPS       | NLL      | CRPS       | NLL       |
> | Vanilla Diffusion | 0.0175     | 2.58     | 0.0157     | -0.96     |
> | ForeDiff          | **0.0173** | **2.30** | **0.0128** | **-1.19** |
>
> Our calibration analysis further supports this interpretation. ForeDiff achieves lower CRPS and NLL than vanilla diffusion, indicating **better-calibrated predictive distributions** rather than collapsed ones. ForeDiff produces narrower yet **better-centered** predictive distributions, simultaneously improving accuracy (e.g., higher PSNR and lower LPIPS) and agreement (lower STD across samples). Taken together, these results confirm that the deterministic predictor provides **structured guidance without enforcing determinism**, enabling ForeDiff to preserve meaningful stochasticity while improving physical plausibility and sampling consistency.

---

> ### Author Response · Authors · 2025-11-19
> **Response to Reviewer bbPz (Part II)**
>
> **[W3] Error propagation from a frozen predictor**
>
> We agree that if the predictive stream were to make large systematic errors, this could negatively affect the final generations. Yet, we believe our design of ForeDiff would potentially mitigate such risks in the following two ways:
> - As mentioned in Sec 3.2, the *deterministic predictor* has **stronger predictive ability** than the *implicit predictor* embedded in a standard diffusion model, a key design of ForeDiff. In other words, compared with vanilla diffusion models, the predictor would be less likely to make large systematic errors. Therefore, we believe that if a predictor were to make large systematic errors, so would a vanilla diffusion model, which **should not be attributed to ForeDiff’s design**.
> - The generative stream is **not forced to follow** a specific predicted future. ForeDiff conditions the DiT branch on the **internal feature representations** of the ViT predictor, rather than its explicit regression outputs. As discussed in **Section 4.3 Analysis** in the *Effect of PredHead module* ablation, we have demonstrated that internal feature representations is more beneficial to the generative stream, indicating its ability to correct the impact of prediction errors.
>
> Taken together, our design trades a weak and entangled *implicit predictor* in vanilla diffusion for a stronger and explicitly trained *deterministic predictor*, and then uses its **features**—not its exact outputs—as condition. This reduces the chance that the generative stream is *driven by faulty premises*, while avoiding hard constraints that would lock the model onto a single predicted trajectory.
>
>
>
> **[Q2] Joint training with a composite loss**
>
> Joint optimization **reintroduces the entanglement** problem that ForeDiff is designed to resolve. When the predictive and generative objectives are optimized simultaneously, the potential learning conflicts would result in a suboptimal information extraction from the predictive stream. This coupling counteracts the core motivation behind our two-stage scheme.
>
> To validate this empirically, we implemented the joint training variant using $\mathcal{L}_{\text{joint}} = \mathcal{L} _ {\text{denoise}} + \lambda \mathcal{L} _ {\text{deter}}$, and trained models on RoboNet with λ=0.1 and λ=1.
>
> |                            | FVD      | PSNR     | SSIM     | LPIPS    | $\text{STD} _ \text{PSNR}$ | $\text{STD} _ \text{SSIM}$ | $\text{STD}_\text{LPIPS}$ |
> | ---- | ---- | ---- | ---- | ---- | ---- | ---- | ---- |
> | Vanilla Diffusion  | 53.8 | 27.1 | 88.2  | 5.65  | 0.66                | 1.33    | 0.65   |
> | ForeDiff-zero    | 52.7  | 27.2  | 88.4  | 5.54  | 0.68  | 1.36  | 0.66 |
> | + joint training ( λ=0.1 ) | 51.6 | 27.3 | 88.6 | 5.40 | 0.66 | 1.29  | 0.63     |
> | + joint training ( λ=1 )   | 52.7 | 27.3 | 88.6 | 5.36 | 0.48 | 0.92   | 0.44 |
> | ForeDiff  | **51.5** | **27.4** | **88.8** | **5.25** | **0.37**  | **0.70**   | **0.35** |
>
> These results reveal a clear pattern. With λ = 0.1, joint training brings only mild gains, but both the overall fidelity and the reduction of sample variance remain noticeably weaker than ForeDiff. Increasing the weight to λ = 1 strengthens variance suppression but begins to harm generative quality, as reflected by the increased FVD—showing that the predictive and generative start to interfere with one another. In contrast, the full ForeDiff design consistently achieves the strongest performance across all metrics, indicating that only the two-stage architectural and training decoupling can simultaneously stabilize predictive representations and support effective denoising.
>
> Overall, the experiments confirm that while joint training can offer mild gains, it ultimately converges to a **compromised middle ground**. The two-stage decoupled design of ForeDiff remains the most effective and stable solution for leveraging deterministic foresight while preserving diffusion’s generative strengths.

---

### Author Response · Authors · 2025-11-28
**Follow-up on author responses**

Dear reviewers and AC,

We would like to kindly check whether any further information is needed from our side regarding the ongoing discussion. We are happy to provide additional clarification if helpful.

Thank you again for your time and consideration.

---

### Author Response · Authors · 2025-12-01
**A Letter to AC: Author Response Summary (Part I)**

Dear AC,

Given that the discussion phase was interrupted and that ACs are now carrying additional responsibility in synthesizing the final decision, we sincerely appreciate your time and effort in managing this process. We also recognize that the lack of reviewer interaction may leave some concerns insufficiently clarified, especially in our case where none of the reviewers have engaged actively in discussion. **As assistance, we would like to provide a concise global summary of how our rebuttal addresses the reviewers’ key concerns**.

Across the four reviews, the main issues cluster around **(1) novelty and relation to prior diffusion practice, (2) sampling consistency vs. predictive accuracy, (3) distributional quality and trade-offs, and (4) fairness and necessity of the two-stage training scheme**. We summarize our clarifications below.

**1. Novelty beyond “common practice” in conditional diffusion**

Reviewer SCh5 and Cbg4 questioned whether ForeDiff resembles common conditioning practices in text-to-image diffusion (e.g., Stable Diffusion). Our response clarifies the fundamental distinction:

- Prior multi-stage diffusion pipelines (e.g., text→image) uses **task-agnostic encoders** pretrained on unrelated objectives, aiming to implement a natural idea for **cross-modal alignment**. These settings do not suffer from the entanglement issue and the diffusion model serves only as a **general representation extraction module**.
- ForeDiff instead introduces a **prediction-aware deterministic pretraining within the same predictive task**, directly targeting a limitation **unique to predictive diffusion**. Specifically, no prior work has addressed the entanglement between condition understanding and target denoising in a single shared backbone.

No reviewer provided references indicating similar decoupling in predictive diffusion, and our literature search found none. Thus, the contribution of ForeDiff is algorithmic, structural, and tailored for predictive learning, **not a reuse of existing conditional-diffusion practice**.


**2. Consistency vs. accuracy: ForeDiff improves both, not one at the cost of the other**

Reviewer SCh5 and bbPz raised concerns that reduced sample variance might indicate mode collapse rather than genuine improvement due to potential misunderstanding. However, our **existing results have already demonstrated this is not the case**:

- ForeDiff improves **mean predictive metrics** (PSNR↑, SSIM↑, LPIPS↓), reflecting higher accuracy.
- ForeDiff simultaneously reduces **sample-to-sample variance**, reflecting stronger consistency.

To further ensure this improvement is not achieved by collapsing uncertainty, we computed the **calibration metrics** (including CRPS, NLL and coverage curves) requested by reviewers for both RoboNet and RT-1. ForeDiff achieves **lower CRPS and NLL** across datasets, indicating **better-centered predictive distributions with more faithful uncertainty calibration**. In coverage analysis, vanilla diffusion’s seemingly “better” coverage stems from **inflated variance**, whereas ForeDiff yields **accurate conditional means with moderate uncertainty**.

Taken together, these results show that ForeDiff improves **both** predictive accuracy and sampling consistency. The reduced variance arises from **more precise conditional alignment**, not from distributional collapse, directly resolving the reviewers’ concern.



**3. Distributional quality (FVD), trade-offs, and headroom**

Reviewer SCh5 noted a slight FVD drop on RT-1. Our rebuttal clarifies that this behavior reflects a **predictive–generative trade-off** rather than a structural weakness:

- FVD reflects **generative realism**, while PSNR/LPIPS reflect **predictive fidelity**; improving one axis can mildly affect the other.
- A deterministic ViT predictor achieves **slightly better mean LPIPS** but suffer **dramatically worse FVD**, showing that collapsing to determinism is not a valid solution.
- As illustrated in **Figure 1**, ForeDiff is designed to **balance** between two extremes: (i) deterministic predictors with strong accuracy but no generative diversity, and (ii) vanilla diffusion models with high diversity but suboptimal predictive coherence.

Importantly, our **additional experiments** indicate that distributional quality has **further headroom**: using an earlier predictive-stream checkpoint (e.g., 0.8M iterations) even improves FVD relative to the default 1.0M configuration. This suggests that the RT-1 behavior arises from a particular configuration choice rather than a limitation of the method. For fairness and reproducibility, we intentionally use a **single unified configuration across all datasets** rather than tuning each dataset individually.

In summary, ForeDiff attains **deterministic-level predictive accuracy** while retaining **diffusion-level generative capability**, and the distributional metrics can be improved further without altering the core method.

---

### Author Response · Authors · 2025-12-01
**A Letter to AC: Author Response Summary (Part II)**

**4. Fairness of compute and necessity of the two-stage scheme**

Several reviewers questioned the added parameters, compute cost, and whether the two-stage procedure is necessary. Our rebuttal provides direct evidence addressing each point:

- **Training fairness.** In training, predictive-stream outputs are **cached**, so stage-2 training does not rerun the predictor. The only extra cost is the one-time stage-1 training.
- **Inference fairness.** At test time, the predictive stream runs **a single forward pass**, while diffusion performs **50 denoising steps**. The added overhead is ≈ **1%**.
- **Parameter-scaling control.** We enlarged the vanilla diffusion model to match ForeDiff’s total depth (18 DiT blocks). As shown in Fig. 8(c), this scaled baseline still underperforms ForeDiff, indicating that the gains arise from **architectural disentanglement**, not increased capacity.
- **Training-only decoupling control.** Pretraining a vanilla diffusion at $t=1$ (Appendix Table 10) modestly reduces variance but **degrades FVD** and remains far below ForeDiff. This shows that **training-scheme decoupling alone is insufficient** without architectural separation.
- **Joint training control ($L_{denoise} + λ L_{deter}$).** With λ=0.1, improvements are minimal; With λ=1, deterministic supervision disrupts denoising and **harms FVD** (Appendix Table 12). Neither setting approaches ForeDiff, demonstrating that **joint optimization reintroduces the entanglement ForeDiff avoids**.

These results collectively show that ForeDiff’s benefits are **architecture-driven rather than compute-driven**, and that the **two-stage decoupling is both essential and empirically validated**.



**5. Other reviewer-specific clarifications (brief)**

All remaining issues have been directly addressed in the rebuttal. Key clarifications include:

- **Deterministic predictor baseline.** Included for all datasets; ForeDiff consistently achieves **much better FVD** and better scientific metrics, showing that deterministic collapse is not a viable alternative.
- **Backbone generality (DiT vs. U-Net / LDM).** ForeDiff is **not DiT-specific**: it only requires access to intermediate predictive features. Our implementation already follows **latent diffusion**, and the same decoupling applies naturally to U-Nets.
- **Baseline breadth.** For video prediction, Table 3 includes strong contemporary baselines (MaskViT, FitVid, iVideoGPT). For scientific forecasting, PDE-based methods (e.g., PINNs, UniSolver) rely on explicit physics, making them incomparable to our data-driven generative setup; even so, their performance is far below larger-scale diffusion models.
- **Scalability / higher-resolution experiment.** Additional RT-1 experiments at **128×128** confirm the **same superiority trend** as at 64×64 (ForeDiff better on all metrics).
- **LPIPS and temporal consistency.** LPIPS in Fig. 3(b) was used only to visualize **intra-sample variability**, following standard practice. Additional temporal-consistency analysis further confirms ForeDiff’s improvement.
- **Text-conditioning.** Text input can be incorporated exactly as in standard latent diffusion—from a text encoder into the predictive stream—since ForeDiff does not modify conditioning mechanisms.
- **Relation to world models.** A world model is a **task formulation**, whereas ForeDiff is a **modeling paradigm**. ForeDiff can serve as a backbone *for* world-model tasks but is not restricted to them.



We hope this summary clearly conveys how our rebuttal addresses all major reviewer concerns, both empirically and conceptually.

Thank you again for your time, especially under the added workload created by the review-system incident. We appreciate your effort and hope this consolidated response assists with your meta-review.

---

### Meta-Review · Area_Chair_LPd1 · 2025-12-30

**Summary:**

This paper aims to improve the sampling consistency in predictive diffusion models by decoupling the predictive learning and the generative denoising learning.

Reviewers commented positively on the novel and well-motivated method, the comprehensive and supportive experimental results, and a well-written paper with good visual presentation.

Reviewers commented negatively on the implementation complexity, weak theory, collapse in deterministic predictor, the potential for more experiments and more metrics, and the modest improvements.

Overall, this is a borderline case, and the strengths slightly outweigh the weaknesses.

**Reviewer Concerns:**

(Weaknesses are indexed using reviewers' original ordering)

For Reviewer bbPz, W1-3 have been addressed.

For Reviewer 6hoQ, W1-5 have been addressed.

For Reviewer SCh5, W1 and W5 have been partially addressed. W2-4 and W6 have been addressed.

For Reviewer Cbg4, W1 has been addressed. W2-3 have been partially addressed.

**Reviewer Scores:**

For Reviewer bbPz, the score is likely to be the same or slightly increased.

For Reviewer 6hoQ, the score is likely to be the same.

For Reviewer SCh5, the score is likely to be the same or slightly increased.

For Reviewer Cbg4, the score is likely to be the same.

---

### Decision · Program_Chairs · 2026-01-26

Accept (Poster)